# Biodegradation of Azo Dye Methyl Red by *Pseudomonas aeruginosa*: Optimization of Process Conditions

**DOI:** 10.3390/ijerph19169962

**Published:** 2022-08-12

**Authors:** Muhammad Ikram, Mohammad Naeem, Muhammad Zahoor, Abdur Rahim, Marlia Mohd Hanafiah, Adeleke Abdulrahman Oyekanmi, Abdul Bari Shah, Mater H. Mahnashi, Amer Al Ali, Naif A. Jalal, Farkad Bantun, Abdul Sadiq

**Affiliations:** 1Department of Chemistry, Abdul Wali Khan University Mardan, Mardan 23200, Pakistan; 2Department of Biochemistry, University of Malakand at Chakdara, Chakdara 18800, Dir Lower Khyber Pakhtunkhwa, Pakistan; 3Department of Zoology, University of Malakand at Chakdara, Chakdara 18800, Dir Lower Khyber Pakhtunkhwa, Pakistan; 4Department of Earth Sciences and Environment, Faculty of Science and Technology, Universiti Kebangsaan Malaysia, Bangi 43600, Selangor Darul Ehsan, Malaysia; 5Centre for Tropical Climate Change System, Institute of Climate Change, Universiti Kebangsaan Malaysia, Bangi 43600, Selangor, Malaysia; 6Division of Applied Life Science (BK21 Plus), Institute of Agriculture and Life Sciences, Gyeongsang National University, Jinju 52828, Korea; 7Department of Pharmaceutical Chemistry, College of Pharmacy, Najran University, Najran 66462, Saudi Arabia; 8Department of Clinical Laboratory Sciences, Faculty of Applied Medical Sciences, University of Bisha, 255, Al Nakhil, Bisha 67714, Saudi Arabia; 9Department of Microbiology, Faculty of Medicine, Umm Al-Qura University, Makkah 24382, Saudi Arabia; 10Department of Pharmacy, Faculty of Biological Sciences, University of Malakand, Chakdara 18800, Dir Lower Khyber Pakhtunkhwa, Pakistan

**Keywords:** biodegradation, methyl red, *P. aeruginosa*, wastewater, textile dyes

## Abstract

Water pollution due to textile dyes is a serious threat to every life form. Bacteria can degrade and detoxify toxic dyes present in textile effluents and wastewater. The present study aimed to evaluate the degradation potential of eleven bacterial strains for azo dye methyl red. The optimum degradation efficiency was obtained using *P. aeruginosa*. It was found from initial screening results that *P. aeruginosa* is the most potent strain with 81.49% degradation activity and hence it was subsequently used in other degradation experiments. To optimize the degradation conditions, a number of experiments were conducted where only one variable was varied at a time and where maximum degradation was observed at 20 ppm dye concentration, 1666.67 mg/L glucose concentration, 666.66 mg/L sodium chloride concentration, pH 9, temperature 40 °C, 1000 mg/L urea concentration, 3 days incubation period, and 66.66 mg/L hydroquinone (redox mediator). The interactive effect of pH, incubation time, temperature, and dye concentration in a second-order quadratic optimization of process conditions was found to further enhance the biodegradation efficiency of *P. aeruginosa* by 88.37%. The metabolites of the aliquot mixture of the optimized conditions were analyzed using Fourier transform infrared (FTIR), GC-MS, proton, and carbon 13 Nuclear Magnetic Resonance (NMR) spectroscopic techniques. FTIR results confirmed the reduction of the azo bond of methyl red. The Gas Chromatography–Mass Spectrometry (GC-MS) results revealed that the degraded dye contains benzoic acid and o-xylene as the predominant constituents. Even benzoic acid was isolated from the silica gel column and identified by ^1^H and ^13^C NMR spectroscopy. These results indicated that *P. aeruginosa* can be utilized as an efficient strain for the detoxification and remediation of industrial wastewater containing methyl red and other azo dyes.

## 1. Introduction

The textile industry is a leading contributor of highly polluting wastewater over the past few decades; the effluents generated from the processing plants are projected to increase on a geometric scale due to industrial growth and expansion. During the last decades ago, the environmental problems arising from the textile industry have gained more attention [1,2]. William Henry Perkin discovered the first synthetic dye in 1856. He named it organic aniline dye (Mauveine) [3]. The textile sector is estimated to produce 80% azo dyes of synthetic origin for dyeing applications annually, with about 10–15% of the dye during processing not binding to the fibers and thus the effluent needs treatment [4]. The aquatic ecosystem is affected by wastewater discharge, which has a direct impact on the health of the fauna and flora. The presence of azo dyes in water disturbs and reduces oxygen solubility and the penetration of light, thus affecting the photosynthetic activity of plants. These dyes also affect water quality and its aesthetic aspects [5,6,7]. The major problem created by the discharge of these effluents is water pollution. Water contamination by these coloring agents seriously endanger human lives, the environment and aquatic life. These dyes have adverse effects due to their chemical structure, which tends to be persistent in nature and they promote various adverse reactions such as angioedema, nasal congestion, itchy skin, hives, hepatic carcinogenicity, and tumors in rats. Moreover, these dyes disturb the composition of aquatic ecosystems and affect the survival of aquatic organisms [8,9,10,11]. These pollutants also cause the synthesis of chlorophyllase and abscisic acid, both of which might result in the breakdown of chlorophyll, which has a negative impact on the chlorophyll content of plants. Aquatic species, particularly fish, are affected by the water quality of their surroundings. Azo dyes and other toxic chemicals cause stunted growth of several fishes and also affect their gills, livers, gills, intestines, and muscles [12,13]. Additionally, these dyes may reduce germination rates and potentially prevent seedling shoot and root extension. Plants have a key role in ecology by producing organic matter that helps to maintain soil fertility, acting as a habitat for species, and preventing soil erosion [9,12]. The dye or its by-products discharged from the textile industry in wastewater pose serious long-term health problems to human beings. Several organs such as kidney, brain, liver, and heart as well as immune, reproductive, and respiratory systems of the human may also be affected. Similarly, diseases such as respiratory problems, allergy, asthma, nausea, dermatitis, eye or skin irritation, and dermatitis are the most common associated health disorders caused by dye effluent when in contact with humans [14]. Discharge of dyes without any proper treatment has the possibility to affect aquatic animals and plants as a result of blockage of sunlight penetration in the receiving water [15]. Therefore, dye-containing wastewater and effluents from textile and other industries must be treated before being released into the environment. The treatment of wastewater is necessary for a sustainable environment [16,17].

Various chemical and physical treatment techniques are used for dye treatment and removal from wastewater and effluents. These techniques include adsorption, photolysis, chemical oxidation and reduction, sonication, chemical precipitation, irradiation, coagulation/flocculation, electrochemical treatment, photochemical process, activated carbon adsorption, ozonation, Fenton process, membrane separations, and ion exchange [18,19,20]. However, there are some limitations in these treatment approaches, such as being unviable commercially and unable to completely remove the azo dyes and their transformation products [21]. Moreover, a large amount of sludge is produced that results in additional pollution and removal of these secondary pollutants requires costly operations [22]. As a result of these challenges, biological techniques which utilize green chemistry are seen as workable substitutes due to their environmental friendliness, cost-effectiveness, ease of operation, and high degradation efficiency [23]. Researchers and industries have become interested in the bioremediation of dye effluents using greener solutions such as microorganisms. Recently, many new approaches for dye biodegradation have been developed which are considered as cost-effective and environmentally friendly. Therefore, biological treatments using microorganisms to degrade synthetic dyes are being investigated as viable and cost-effective alternatives as compared with other treatment approaches [24,25].

The literature demonstrates that utilizing bacteria as a precursor for the bioremediation of azo dyes has demonstrated better results in the most diverse procedures and conditions [16,26]. The bacterial strains have the ability to degrade and transform many toxic compounds into non-toxic end products [27]. Different bacterial species have been reported for decolorization and degradation studies of dyes in the literature; some of these are given in Table 1.

Bacteria have enzymes as such oxidoreductive, veratryl alcohol oxidase, azoreductase, laccase, and peroxidase, which allow them to break down the dyes in industrial wastewater and effluents [41,42]. Bacterial degradation of azo dyes in textile and tanning effluents usually involves two steps. In the first step, the azo linkage of the dye is reduced through the azoreductase enzyme, resulting in aromatic amines formation. These products (aromatic amines) are toxic in nature; they are further degraded in the second step into less toxic metabolites or are completely mineralized [43,44]. The degradation mechanism of other azo dyes brought about by *P. aeruginosa* has been reported in the literature. Recently, Ullah et al. has reported an insight into the mechanism of biodegradation of other azo dye methyl orange by *P. aeruginosa* [32].

However, to date, only limited studies are available in the literature on the biodegradation of methyl red and degradation mechanisms especially using *P. aeruginosa*. The aim of the present study was to investigate the degradation efficiency and mechanism of *P. aeruginosa* out of the available strains for methyl red dye. The impacts of several physicochemical conditions were also investigated for better degradation of methyl red. Conditions in a one-factor-at-a-time process and the interaction of the process variables were optimized and analyzed using the central composite design (CCD) of response surface methodology (RSM). The extracted isolates were then subjected to column chromatography to obtain metabolites in a pure state, which were then subjected to structural analysis using spectroscopic techniques that include the FTIR, GCMS, and NMR.

## 2. Materials and Methods

### 2.1. Dye and Other Reagents

The textile azo dye methyl red was provided by a textile mill located in Karachi, Sindh Pakistan. Nutrient broth, sodium chloride, glucose, n-hexane, ethyl acetate, hydrochloric acid, redox mediators, sodium hydroxide, and other chemical reagents were of high analytical grade and quality. The other chemical reagents were purchased from Sigma Aldrich, Germany.

### 2.2. Bacterial Strains

Biodegradation efficiency of *Pseudomonas aeruginosa* (ATCC 27853), *Staphylococcus aureus* (ATCC 27700), *Citrobacter amalonaticus* (ATCC 25405), *Escherichia coli* (ATCC 25922), *Staphylococcus epidermidis* (ATCC 14990), *Proteus mirabilis* (ATCC-29906), *Enterobacter sakazakii* (ATCC 29544), *Xanthomonas campestris* (ATCC 13951), *Bacillus subtilis* (ATCC 6633), *Streptococcus pyogenes* (ATCC 12344), and *Salmonella enterica* (ATCC 43971) were tested using methyl red dye. The above-mentioned and identified cultures were obtained from the Department of Biotechnology, University of Malakand and Department of Microbiology, Abdul Wali Khan University Mardan, Pakistan.

### 2.3. Preparation of Dye Solution

40 ppm of dye stock solution was prepared by dissolving 0.04 g of dye dye in little amount of distilled water in conical flask and then the volume was reached to 1000 mL. Then, the mixture was continuously agitated for 5 min to ensure homogenous mixing; hence 40 ppm concentration of the aqueous stock solution was prepared and stored at room temperature for the other degradation experiments.

### 2.4. Growth Media and Culture Growth

Bacteria need growth media for survival and proper growth. Therefore, broth solution was prepared by dissolving 13 gm nutrient broth (Sigma Aldrich, Burlington, MA, USA) in distilled water until the volume of 1000 mL was reached. Growth media, test tubes, conical flasks, and all other glassware were sterilized in an autoclave machine at 121 °C for the total run of 3 h. When the temperature dropped from 121 °C to 40 °C; the door of autoclave was opened and nutrient broth solution and test tubes were removed from it. Then, to inoculate bacteria in the medium, nutrient broth solution and test tubes were transferred from the autoclave (A55 Autoclave Robust Technology) and were kept inside the laminar flow hood. Each test tube was marked, and then 10 mL of nutritional broth solution was added, followed by the inoculation of bacterial culture. To ensure bacterial growth, the test tubes after inoculation were incubated in an incubator (Lab Hot air Oven DIGI System Laboratory Instrument System and Corporation, New Taipei City, Taiwan) for 24 h at 37 °C.

### 2.5. Degradation/Decolorization Assay

After 24 h when bacterial culture was fully grown, each test tube was filled with 5 mL methyl red solution. Prior to degradation, the absorbance of the supernatant was measured at different time intervals in the visible region of a UV-Visible spectrophotometer (UV-1800 ENG.SOFT) at the dye’s absorption maximum wavelength (430 nm). To separate the bacterial cell mass, aliquots (5 mL) of the degraded mixture were centrifuged for 10 min at 10,000 rpm after 3 days. The bacterial cell mass was separated from the supernatant and the supernatant was subsequently used to conduct the decolorization analysis. The initial and final absorbance values were recorded to compute the percent degradation according to Equation (1) given below [32,45].
(1)$$\%\mathrm{Degradation}/\mathrm{Decolorization}=\frac{\mathrm{Initial}\;\mathrm{absorbance}-\mathrm{final}\;\mathrm{absorbance}}{\mathrm{Initial}\;\mathrm{absorbance}}\times 100$$

The bacterial strain with the highest percentage degradation/decolorization was used for further decolorization/degradation studies.

### 2.6. Optimization of Physiochemical Parameters and Response Surface Optimization of Most Significant Parameters

Physiochemical parameters such as dye concentration, temperature, sodium chloride concentration, time, pH, urea concentration, glucose concentration, and redox mediators were tested in separate experiments to obtain optimum degradation conditions for methyl red dye. In each case, average mean value was calculated by conducting experiments in triplicates. To determine the effect of parameters and the interaction of process conditions to maximize the degradation efficiency by *P. aeruginosa* for the degradation of methyl red, a one-at-a-time factor optimization was conducted on the optimum bacterial strain. The most significant factors influencing higher degradation efficiency were optimized using a central composite design (CCD) of the response surface methodology (RSM), which was conducted on the optimum bacterial strain that had better degradation capacity of methyl red.

The evaluation of the quadratic coefficient of the experimental outcome was evaluated based on the agreement between the predicted values and observed values as indicated in Equation (2).
(2)$$Y=\beta_{0}+\sum_{I=1}^{K}\beta_{I}X_{i}+\sum_{I=1}^{K}\beta_{II}X_{i}{}^{2}+\sum_{I>j}^{K}\beta_{Ij}X_{1}X_{j}+€$$
where *β*_0_ and *β_i_* demonstrate the constant coefficient, while *X_i_* and *β_ii_* represent the linear coefficient of the input parameters and the quadratic coefficient of the input parameters respectively. Similarly, *β_ij_* indicates the interaction coefficient between the input parameters *X_i_* and *X_j_* and € signifies the error parameter of the model.

#### 2.6.1. Dye Concentration Impact on Biodegradation

To investigate the impact of concentration of dye on degradation; the selected strain was cultured in eight test tubes having nutrient broth (10 mL) for 24 h. After culture growth; 5 mL methyl red solution was added to the test tubes at concentration of 5, 10, 15, 20, 25, 30, 35 and 40 ppm respectively. A total of eight solutions(control) comprising 10 mL nutritional broth and 5 mL dye solution were also made for each concentration. After 3 days, the culture mixture containing the dye degraded products was centrifuged in centrifuge machine at 10,000 rpm for 10 min at room temperature. The sample was filtered through 0.2 μm-sized filter paper. Using a UV-Visible spectrophotometer, the final absorbance value of the extracted supernatant mixture obtained after centrifugation was measured at 430 nm [46].

#### 2.6.2. pH Effect on Dye Biodegradation

Bacteria can survive and grow at suitable pH; therefore, sterile nutrient broth was added in fourteen test tubes and inoculated with bacterium *P. aeruginosa* and then incubated at 37 °C in an incubator. When the bacterial culture was fully grown after 24 h; 5 mL dye solution from stock solution was added to each test tube. Control solutions were also prepared as references. The pH values in control reference solutions and inoculated tubes were adjusted by 1 M NaOH and 1 M HCl solution. By adding minute quantity of acid or base on micropipette in respective tubes; the pH values were adjusted using pH indicator strips (Merck KGaA Darmstadt Germany). The supernatant obtained after centrifugation was filtered through filter paper (0.2 μm). The % decolorization of supernatant withdrawn after 3 days was measured by the above-mentioned method using a UV-Visible spectrophotometer.

#### 2.6.3. Temperature Impact on Dye Biodegradation

To investigate the effect of temperature on the degradation of methyl red, 10 mL of solution was added to six test tubes and inoculated with the selected culture. After bacterial growth five mL of methyl red solution was also added to each test tube from methyl red stock solution (40 ppm). Control solution having 5 mL methyl red and 10 mL nutrient broth was also prepared. Test tubes were incubated at 25, 30, 35, 40, 45, and 50 °C in the incubator. After a three-day interval, the degraded samples in the test tubes were spun in centrifuge machine for 10 min at 10,000 rpm, using 0.2 μm-size filter paper, after which each sample was filtered. The % decolorization was determined in the same manner as described previously.

#### 2.6.4. Glucose Effect on Dye Biodegradation

The main source of energy is glucose for bacteria, which also serves as a carbon source (provide carbon atom). Bacterial culture was grown in nutrient broth media in test tubes and incubated at 37 °C, after 24 h 5 mL dye solution was added to culture test tubes and different concentration of glucose; 333.33, 666.66, 1000, 1333.33, and 1666.67 mg/L, were added to each inoculated test tube containing the dye solution. Control solutions containing 5 mL of dye solution and 10 mL of media were also prepared for each concentration of glucose. The degraded sample after centrifugation and filtration through 0.2 μm filter paper was subjected to UV-Visible spectrophotometry. The percentage decolorization of the supernatant was measured by same method as described in Equation (1).

#### 2.6.5. Sodium Chloride Effect on Dye Biodegradation

The dye degradation activity is affected by sodium chloride salt because it is a major salt that increases sea water salinity. Degradation of pollutants and dyes usually occurs at optimum saline conditions. Dye solution (5 mL) was added to five test tubes having the cultivated culture of *P. aeruginosa.* Different concentrations such as 333.33, 666.66, 1000, 1333.33 and 1666.67 mg/L of sodium chloride were also added to each inoculated test tube. Reference solutions were also prepared for each concentration of sodium chloride. The supernatant withdrawn after centrifugation was used to determine its percentage decolorization.

#### 2.6.6. Impact of Incubation Time on Dye Biodegradation

Nutrient broth in amount of 30 mL was taken in large size test tube; inoculated with the culture of bacteria and incubated for 24 h. After culture growth 15 mL dye solution was added to it. Percentage degradation of the dye was monitored after 1 day interval till 6 days. After 6 days, due to no significant increase, the percent degradation was then recorded after 3 days interval till to 21 days. Control solution having 5 mL dye and 10 mL media was also prepared as reference. The percentage degradation rate was measured after every 3 days interval till 21 days using UV Visible Spectrophotometer.

#### 2.6.7. Urea Impact on Dye Biodegradation

Urea serves as a nitrogen source for bacterial growth, so a substantial amount of urea is required for bacteria to survive and grow. The selected culture of bacteria was cultured in five test tubes containing 10 mL nutrient broth for 24 h. When bacterial growth was fully ensured after 24 h, each test tube was amended with 5 mL methyl red dye solution. 333.33, 666.66, 1000, 1333.33 and 1666.67 mg/L of urea concentration was introduced to inoculated test tubes containing methyl red solution. Control dye solutions were also prepared for each concentration. The percentage (decolorization/degradation) of supernatant was determined by UV-Visible spectrophotometry at three-day intervals from initial and final absorbance values.

#### 2.6.8. Redox Mediators Impact on Biodegradation of Methyl Red

Redox mediators such as uric acid, ethylenediamine tetra acetic acid, sodium benzoate, and hydroquinone were used in this study. Bacterial culture was cultivated in four test tubes and methyl red solution was added after culture growth to each test tube. Uric acid, hydroquinone, ethylenediamine tetra acetic acid, and sodium benzoate at a concentration of 66.66 mg/L were added to each test tube. Four uninoculated control solutions or reference solutions were also used for each concentration of redox mediators. The supernatant withdrawn after centrifugation and filtration was subjected to UV Visible spectrophotometry. The absorbance values were recorded. After determining the percentage absorbance by formula, the redox mediator having the highest percent degradation value was selected and used in the biodegradation study.

### 2.7. Biodegradation of Methyl Red at Optimal Conditions

The impacts of various physiochemical parameters such as dye, glucose, temperature, sodium chloride, time, redox mediator, and urea were assessed in order to determine the best optimal conditions for degradation by *P. aeruginosa*. Following the determination of these ideal parameters, a degradation experiment was carried out using these conditions at the final stage of the single experimental study. The optimization of degradation efficiency was investigated based on the interaction of process conditions using the central composite design (CCD) of the response surface methodology (RSM).

### 2.8. Metabolites Extraction and Isolation after Biodegradation of Methyl Red

The mixture of bacterial mass and degraded dye at optimal conditions was homogenized and centrifuged for 10 min in centrifuge at 10,000 rpm and then filtered. Cell-free culture supernatant was later used for metabolites extraction. Ethyl acetate was mixed with the supernatant in equal volume and then agitated vigorously for 25 min. A separating funnel was used for the separation of organic or ethyl acetate and aqueous phases. The evaporation of ethyl acetate at 40 °C was ensured to obtain extract of the degraded products in solid state.

#### 2.8.1. Metabolites Isolation and Purification through Silica Gel Column

A silica gel column was used to separate the metabolites that were extracted in ethyl acetate. Slurry was prepared by mixing a small amount of the extract with silica gel. The slurry was then dried with air to remove any remaining solvent or moisture. A column of 50–100 cm height and diameter of 4 cm was filled with silica gel. After that, n-hexane solvent was used to wash the column. The crude extract in the form of slurry was subjected to column at the top, washed with n-hexane, and then eluted with several solvent systems containing varied ratios of ethyl acetate and n-hexane (1:2, 4:1, 5:1, 1:1, and 1:5). The fractions from the silica gel column were collected in glass vials with 5 mL volume.

#### 2.8.2. Thin Layer Chromatography (TLC) Profiling

Based on similar Rf values and thin-layer chromatography (TLC) profiling, the similar fractions were recombined. The developing solvent system used was consist of ethyl acetate and n-hexane (2:3) was used. The spots on TLC plates were observed under ultra violet (UV) light. The metabolites were confirmed by TLC. Only one fraction of the metabolite was identified using TLC profiling.

### 2.9. Metabolites Analysis by Gas Chromatography Mass Spectrometry (GC-MS)

Identification of the metabolites formed after biodegradation was performed by an Agilent USB-393752 gas chromatograph (Agilent Technologies, Palo Alto, CA, USA) with a HHP-5 MS 5% phenyl methyl siloxane capillary column (30 m × 0.25 mm × 0.25 μm film thickness; Restek, Bellefonte, PA, USA) equipped with an FID detector. The oven was first kept at 70 °C for 1 min, then the temperature was raised to 180 °C within 5 min, and finally the machine’s temperature was raised to 280 °C for 20 min. The temperature difference between the injector and detector was 220 °C. Following the injection of the sample in a split-less mode in an amount of 1 L, helium was employed as the carrier gas at a flow rate of 1 mL/min. The same operating conditions as before were applied to the GC-MS analysis of the metabolites utilizing an Agilent HP-5973 (Ramsey, Minneapolis, MN, USA) mass selective detector in electron impact mode (ionization energy: 70 eV). The retention times of identified metabolites were compared with those of the reported compounds in the literature.

### 2.10. Analysis of Methyl Red Metabolites by Fourier Transform Infrared (FT-IR) and Nuclear Magnetic Resonance (NMR) Spectroscopy

The Perkin Elmer Spectrum Two instrument (103385; Waltham, MA, USA) was used to conduct the FTIR analysis of the methyl red before and after bacterial treatment while the purified fraction obtained through silica gel column after TLC profiling was analyzed by Proton (^1^H) and Carbon 13 (^13^C) NMR (Advance III 600 MHZ, Cryoprobe, Triple Resonance Channel; Bruker, Billerica, MA, USA) for confirmation and structure elucidation.

### 2.11. Statistical Data Analysis

All the experiments including control groups through out the experiments were repeated in triplicates, and the results obtained were presented as the Mean ± Standard deviation. The effect of the process conditions and the interaction for the evaluation of optimum degrdation efficiency of bacteria was analyzed according to ANOVA at *p* < 0.05. The optimization of degradation efficiency was investigated based on the interaction of process conditions using the central composite design (CCD) of the response surface methodology (RSM).

## 3. Results and Discussions

### 3.1. Most and Highly Potential Bacacterial Strain for Biodegradation of Methyl Red

The degradation potential of different bacteria is different. Out of 11 bacterial strains, *Pseudomonas aeruginosa* emerged as the most effective strain for degradation of the methyl red. Its percent degradation potential was 81.49%. The percent degradation of other bacteria is given below in Figure 1. Hence, the effect of operational conditions was investigated on the optimum biodegradation capacity of the bacteria strain which was found to be *P. aeruginosa* in a one-at-a-time factor optimization. Furthermore, the optimization of the effect of the interaction of operational factors of the process conditions of the most significant parameters was conducted using CCD of the RSM.

### 3.2. Methyl Red Degradation at Optimum Conditions

#### 3.2.1. Dye Concentration Effect on Biodegradation

Concentration of dye affects the degradation potential of bacteria. In Figure 2, the effect of concentration of methyl red on percentage degradation of dye is given. It is clear from the result that the highest degradation rate (68.59%) was observed at 20 ppm dye concentration; since dye is a chemical substance. its higher concentration may inhibit effective interaction between dye molecules and the precursor thereby limited the bacterial efficiency for dye degradation. Zhuang et al. claimed that high concentrations of dye due to toxicity block the active sites of bacterial enzyme, thus having the tendency to inhibit the bacterial potential for dye degradation [47]. Therefore, it can be concluded that degradation rate of dye increases when the concentration of dye decreases and vice versa.

#### 3.2.2. Impact of pH on Dye Biodegradation

pH is one of the most important parameters affecting the activity of enzyme and degradation potential of the bacteria. In Figure 3, the pH impact on methyl red degradation is shown. Increase in degradation was observed when the pH increased from the acidic to the alkaline region. At pH 9, the decolorization rate increased (62.32%) and was found to decrease at low pH, which indicates that bacterial growth and enzymatic activity is affected at extreme alkaline and acidic conditions. According to Shi et al., the pH range for bacterial biodegradation commonly optimizes from 6 to 10 [48]. Ikram et al. [49] found that the biodegradation rate at pH 10 is the maximum. Mostly, the decolorization of dyes is usually achieved under alkaline conditions at pH ranges of 6 to 10 [50].

#### 3.2.3. Impact of Temperature on Biodegradation of Dye

Temperature affects the biodegradation activity of bacteria. In Figure 4, the effect of temperature on dye degradation is shown. Temperature affects the growth of bacteria, and consequently the biodegradation potential of bacteria decreases. At 40 °C, the highest decolorization (64.69%) was noted, which implies that above or below this temperature, the decolorization potential of *P. aeruginosa* decreases due to slow growth of the culture. Anjaneya et al. [51] reported that bacterial enzymes become inactive at high temperatures, and consequently the decolorization rate of bacteria considerably decreases. Pearce et al. [52] reported that the optimum growth temperature for the degradation of reactive azo dyes ranges from 35 to 45 °C until the maximum degradation of dye is achieved. Above the optimum temperature, a steady decrease in decolorization occurs, which may be due to the denaturation of bacterial enzymes. Maximum decolorization of Red dye I by Rhodobecter *sphaeroides* has been reported at 35–40 °C [53]. According to Bhatt et al., *P. aeruginosa* NBAR12’s color removal efficiency increased up to 40 °C with its peak activity; subsequent increases led to a decrease in decolorization potential [54].

#### 3.2.4. Incubation Time and Its Impact on Dye Biodegradation

Time also affects the degradation activity of bacteria. In Figure 5, the impact of time on methyl red degradation by bacterium *P. aeruginosa* is given. The degradation of the dye was monitored after 1 day interval till 6 days. After 6 days, due to no significant increase, the percent degradation was recorded every 3 days for up to 21 days. After 3 days of incubation a maximum in degradation was observed. No significant increase in degradation was seen after these 3 days. This is most likely due to reaching the stationery and death phase. Similar findings were reported by Ikram et al., with the biodegradation of another azo dye, Basic Orange 2, by *E. coli* throughout the course of three days of experimentation, with the highest % degradation being attained. The maximum decolorization has been noted after 3 days interval after this no significant increase in degradation was observed [55]. 

#### 3.2.5. Impact of Glucose Concentration on Dye Biodegradation

A substantial amount of glucose is required for bacteria as it is the main source of energy and also acts as a carbon source. Nevertheless, some dyes are complex in nature and are difficult to degrade. Therefore, an additional source is required to supplement carbon [56]. The rate of decolorization is positively impacted by an increase in glucose concentration, but, after a certain point, degradation activity declines, possibly as a result of a negative impact on the bacterial metabolic pathway’s ability to catabolize sugar [57]. Figure 6 illustrates the impact of glucose on the dye degradation. The addition of 1666.67 mg/L glucose showed a high degradation rate (64.45%). The lowest percent degradation (52.03%) was observed at 333.33 mg/L. According to the published reports, the most efficient and convenient carbon source for the microbial degradation of dyes or dye intermediates is glucose [58].

#### 3.2.6. Urea Concentration Effect on Dye Biodegradation

Bacteria utilize urea as a source of nitrogen, so a significant amount of urea is needed for bacteria to break down the given dye. Figure 7 shows the influence of urea concentration on the percentage degradation of the methyl red. The selected dye revealed a high rate of degradation (67.99%) at 1000 mg/L. The breakdown activity reduced as the content of urea increased due to induced toxicity. According to Ikram et al. [55], at a concentration of 1000 mg/L of urea, the Basic Orange 2 dye demonstrated an accelerated rate of degradation of 80.92 percent. Due to urea toxicity and the increasing urea content, the percentage breakdown activity reduced at higher concentration to 59.59 percent.

#### 3.2.7. Impact of Sodium Chloride Concentration on Biodegradation

The impact of sodium chloride concentration degradation of methyl red by selected bacterial strain is depicted in Figure 8. It is revealed that as the concentration increases, the ability of bacteria to degrade dye decreases. The highest percentage degradation (61.75%) was found at 666.66 mg/L salt concentration for the degradation of methyl red by *P. aeruginosa*. High salt concentrations result in bacterial cells plasmolysis, which reduces the growth of bacteria and consequently inhibits the degradation capability of bacteria [59]. Asad et al. also reported that biodegradation of Dye Brown dye was enhanced at low concentrations of sodium chloride supplementation [60]. Similar results were recorded by Ikram et al.; the percentage degradation of Basic Orange 2 by *E. coli* at a concentration of 666 mg/L sodium chloride was 82.35; the results indicated that a low concentration of salt is required for bacteria effectively degrade the dye [55]. Joe et al. studied the degradation of Ramazol Black B *P. aeruginosa* CR-25 at varied salt (NaCl) concentrations (0%, 0.5%, 2%, 4%, 5%, 6%) and reported that *P. aeruginosa* CR-25 was able to decolorize the dye at 0.5%, 2%, and 4% salt concentrations; the decolorizations of RBB were 66%, 65%, and 63% respectively after 48 h of incubation. However, when the concentration of salt increased to 5% and 6%, the decolorization activity was affected; there was no increase in the decolorization [61]. Therefore, it can be concluded that a sufficient amount of salt is needed for bacteria to decolorize the dye, while higher concentrations affect the bacterial decolorization activity.

#### 3.2.8. Redox Mediators Effect on Dye Degradation

Redox mediators function as electron donors and acceptors. These are used by bacteria during the electron transfer process. An oxidation-reduction reaction results in dye decolorization. Bacterial strains degrade the azo bond of the dyes either aerobically or in anaerobic conditions. Figure 9 shows redox mediators’ effect at a concentration of 66.66 mg/L on dye biodegradation. It was revealed that hydroquinone influenced higher degradation capability (62.08%) of the methyl red by bacterial strain. It has been reported that quinones and hydroquinone are very helpful in the reduction of azo bond of dye molecule [62]. Our findings are very similar to those of literature reported study in which hydroquinone was considered as best choice for use as redox mediator. In other studies, adding hydroquinone as a mediator increased the yield after 5 h of incubation from 79.35% in the control degradation study (dye in nutritious broth without redox mediator) to 94.41% [63]. As a result, adding quinones as redox mediator improves the flow of electrons from the electron donor to the electron acceptor, which is often an azo dye. Typically, this quicker transfer results in a higher rate of color removal [64,65]. The electron transfer phase can be accelerated with supplementation of small amount of redox mediator while lowering dye molecule steric hindrance [66,67].

### 3.3. Methyl Red Degradation at Optimum Physio-Chemical Conditions

After confirming the effects of various physio-chemical conditions on the biodegradation of dye by a given bacteria, such as the concentration of dye (methyl red), redox mediators, sugar concentration, urea concentration, pH, salt concentration, time, and temperature; the degradation of methyl red was conducted by employing these optimal conditions in a single-factor optimization of experimental analysis. The optimum conditions found were 20 ppm methyl red concentration, pH 9, temperature 40 °C, 1666.67 mg/L glucose concentration, 666.66 mg/L sodium chloride concentration, 1000 mg/L urea concentration, 3 days incubation duration, and 66.66 mg/L hydroquinone (redox mediator). The other experimental parameter conditions were the same as those described earlier. Consequently, the percentage degradation reached to 88.37%; when these optimal physiochemical conditions were employed in a single experiment. Figure 10a,b shows methyl red treatment before and after *P. aeruginosa* treatment at the optimum physiochemical conditions. There were significant visual changes in color and decolorization after *P. aeruginosa* treatment, which indicated that the bacterium degraded the dye and some new products or metabolites formed.

### 3.4. Most Significant Parameters and Their Response Surface Optimization

The interactive effect of process factors was conducted using the response surface plot of the central composite using design expert 6.0.4 software.

The effect of the interaction and prediction of the relationship between the predicted and observed values was evaluated by the second-order quadratic function of the response surface. The experimental conditions for the optimization of biodegradation of methyl red were designed according to Table 2. A close agreement was found between the observed values and the predicted values according to the experimental design in Table 3, which indicated the aptness and suitability of the model for the prediction of biodegradation of methyl red dye using *P. aeruginosa*. The significance of the effect of the interaction of the process conditions indicating the relationship between the operational factors is illustrated according to Equation (3).
(3)$$Y=65.95+3.70X_{1}+1.05X_{2}+1.58X_{3}+9.39X_{4}-3.94X_{1}{}^{2}+2.21X_{2}{}^{2}+0.60X_{3}{}^{2}\;-4.84X_{4}{}^{2}+3.37X_{1}X_{2}-0.22X_{1}X_{3}-1.23X_{1}X_{4}-1.31X_{2}X_{3}+0.89X_{2}X_{4}\;-2.15X_{3}X_{4}$$

The statistical quantitative analysis of the prediction of percent biodegradation of the dye using the bacteria strain was evaluated using analysis of variance (ANOVA). It was revealed from the assessment that the optimum biodegradation of methyl red was achieved by 80.23%. This indicated that the effect of interaction of incubation time, pH, dye concentration, and temperature enhanced the degradation efficiency of *P. aeruginosa*. This was achieved at a low probability coefficient (*p* < 0.05) and high F-Value. Similarly, a high value of Adequate precision (Adeq) precision, a value of 7.931, was achieved, which was higher than the value of 4 (Table 4). Generally, the prediction of the model is sound, effective, and suitable if the value of Adeq precision is greater than 4 [68,69]. It can be revealed that pH and incubation time have significant impacts on the degradation of the dye. The synergistic interaction of the effect of the process conditions according to the experimental design for the optimal degradation efficiency of *P. aeruginosa* is presented in Figure 11a–d. The degradation capacity of bacteria strains greatly depends on pH, microbial growth, and the conversion of complex substances into a simpler form [70]. The result indicated that pH 9 represents the best environmental condition at increased incubation time for the biodegradation of methyl red (Figure 11a). When incubation time increase the pH facilitates the interaction of methyl red to the cell membrane of the bacterial strain, thereby resulting in increase in the dye degradation. The cell membrane permeation efficiency may be affected by, it has been identified as the limiting step in the breakdown of dye by bacteria [71]. Meanwhile, the effect of the interaction of pH and temperature had no significant difference on the degradation efficiency of *Pseudomonas*
*aeruginosa* on methyl red dye (Figure 11b), although the effect of temperature in the interaction with pH at 34.55 °C enhanced the degradation of the dye. The interaction between incubation time and dye concentration (Figure 11c), and also the interaction between temperature and dye concentration, as shown in Figure 11d, influenced the degradation of methyl red using the bacterial strain. However, with the consideration of the four-process condition, optimal degradation was enhanced when the pH was 10, the incubation period was 19.4 days, the temperature was 31.54 °C, and the dye concentration was 17.24 ppm.

### 3.5. Characterization Study

The characterization of metabolites for the biodegradation of methyl red dye was analyzed by using Fourier Transform Infrared (FT-IR), Gas Chromatography–Mass Spectrometry (GC-MS), Thin Layer Chromatography (TLC) Analysis, and Nuclear Magnetic Resonance (NMR) Spectroscopy.

#### 3.5.1. Fourier-Transform Infrared (FTIR) Analysis

The methyl red and degraded products (metabolites) of optimal conditions experiment were analyzed/characterized by Fourier-Transform Infrared (FTIR). FTIR spectrum of the original undegraded dye and the degraded dye were compared. According to FTIR spectra, certain functional groups of the original dye molecule were not present in the dye-degraded products, and some new peaks appeared. FTIR spectra of methyl red dye before bacteria treatment are given in Figure 12a. The peak at 3318 cm^−1^ represents an N-H stretch, while the peak at 3185.51 cm^−1^ indicates an O-H stretch. The peak at 2968.79 cm^−1^ is stretching for O-H. The peak at 1587.52 cm^−1^ represents an N=N stretch or azo bond stretch. The peak at 1157.30 cm^−1^ is for an amine C-N stretch. At 821.38 cm^−1^ there is a peak for the C-H stretch of the benzene ring. FTIR spectra of methyl red after bacterial treatment are also clear from Figure 12b. The peaks at 3387.80 cm^−1^ represent amines N-H stretch and 2929.21 cm^−1^ for the O-H stretch appeared and the stretch of 1587.52 cm^−1^ for N=N was found to disappear, which clearly indicated that the azo bond was reduced and the dye was degraded by bacteria. The peaks at 1453.70 cm^−1^ and 1688.17 cm^−1^ are C-H stretches. Moreover, the peak at 1269.04 cm^−1^ represents the C-O stretch of the carboxylic acid group attached to the benzene ring, while the peak at 706.59 cm^−1^ shows C=C stretching of the aromatic benzene. By comparing the FTIR spectra of the original dye to those of the degraded dye or metabolites, significant differences can be noticed. After degradation, some new peaks appeared, while old peaks of the methyl red dye were found to disappear, which indicated that the dye had transformed to new compounds or metabolites. FTIR study confirmed and contributed in the disappearance and reduction of the azo linkage of the methyl red dye. 

#### 3.5.2. Gas Chromatography and Mass Spectrometry (GCMS) Analysis

Figure 13a–c show GC and GCMS chromatograms of dye-degraded products or metabolites. The concentration of metabolites was calculated by comparing the peak area and retention times in the sample with the peak area of the standard reported compounds in literature. The dye metabolites found at RT 9.58 min and RT 2.24 min with a charge-to-ion mass of m/z that closely relate to the methyl red structure were identified as benzoic acid and ortho-xylene respectively. In Table 5, the details of two major metabolites obtained by GCMS results are presented. The GCMS results revealed the confirmation of two important metabolites, i.e., benzoic acid and ortho xylene. The azo bond of the dye was broken by bacterial azoreductase enzyme, leading to the formation of the 2-aminobenzoic acid ring and N,N-dimethylaminobenzene. By subsequent enzymatic actions, 2-aminobenzoic acid converted to benzoic acid (m/z 122) while N,N-dimethylaminobenzene to ortho xylene(m/z 106).

#### 3.5.3. Thin Layer Chromatography (TLC) Analysis

The resultant metabolites extracted in ethyl acetate were passed through silica gel column for purification and isolation. The different fractions obtained were subjected to TLC analysis. Based on similar Rf values and thin-layer chromatography (TLC) profiling, the similar fractions were recombined. The TLC profiling under UV light observation showed that three spots of degraded mixture covered the same distance, which indicated that only one metabolite was confirmed as shown in Figure 14. These fractions were recombined and tested in UV light. The distance covered by the original dye and metabolite on TLC plates observed under UV light was totally different from each other The TLC results confirmed only one metabolite obtained in a pure state in silica gel column fractions, the structure of which was then confirmed by ^1^H NMR and ^13^C NMR spectroscopy.

#### 3.5.4. NMR Spectral Analysis of Methyl Red Biodegradation

The purified fraction obtained through silica gel column after TLC profiling was subjected to Proton (^1^H) and Carbon 13 (^13^C) NMR analysis for confirmation and structure elucidation. ^1^H NMR and carbon 13 NMR spectra of Orange methyl red dye are shown in Figure 15 and Figure 16 respectively. By comparing the ^1^H and carbon 13 NMR spectra of the original methyl red dye to that of the metabolites formed through biodegradation by *P. aeruginosa*, significant changes were found. Out of these metabolites, only one metabolite was obtained in a pure state in silica gel column fractions. ^1^H and carbon 13 NMR spectra of the isolated fraction are shown below in Figure 17 and Figure 18 respectively.

##### NMR Spectral Analysis of the Isolated Metabolite

The metabolite isolated through the silica gel column after TLC profiling was subjected to ^1^H-NMR and ^13^C-NMR analysis. Figure 17 and Figure 18 shows the ^1^H-NMR and ^13^C-NMR spectra of an isolated metabolite. Figure 18 shows peaks at δ 133.29, δ 131.23, δ 129.72, and δ 129.00 ppm respectively, indicating the signals from six carbon atoms on the aromatic ring. The peak at δ 167.80 ppm is a carbon atom of the carboxylic acid group (-COOH). The signal peak at 40.54 represents methyl carbon of DMSO. As shown in Figure 17, the ^1^H-NMR analysis showed three peaks at δ 7.591, δ 7.631, and δ 7.970 ppm, thus indicating signals from five hydrogen atoms. The signal at δ 2.516 ppm represents a singlet proton peak of the carboxylic acid group (-COOH). Thus, a chemical compound with the molecular formula C_7_H_6_O_2_ (benzoic acid) formed if the ^1^H-NMR and ^13^C-NMR data analyzed in Figure 17 and Figure 18 were combined together. Figure 19 shows the chemical structure of an isolated metabolite (benzoic acid).

### 3.6. Proposed Biodegradation Pathway for the Biodegradation of Methyl Red by P. aeruginosa

The results of FTIR, GC-MS, and NMR indicated that the molecule of the methyl red dye was broken down by *P. aeruginosa* and new products or metabolites were formed. A number of bacterial species such as *Enterococcus faecalis YZ 6, Pseudomonas* sp., and *Bacillus* sp. have azoreductase enzymes [72,73,74,75]. The azo linkage (–N=N– bond) of methyl red was broken by the enzyme azoreductase enzyme of *P. aeruginosa* and converted to two substituted benzene derivatives, dimethylaminobenzene and 1-amino benzoic acid. Furthermore, 1-amino benzoic acid was converted to benzoic acid through deamination process by deaminase enzymatic action. The deaminase enzyme of the bacterial system influences the conversion of N,N-dimethylaminobenzene to N,N-dimethyl benzene by deamination reaction. After that, the reductive demethylation of N,N-dimethyl benzene possibly could form an aminobenzene ring. The aminobenzene molecule undergoes deamination by the bacterial deaminase enzyme, leading to the formation of benzene. Methyltetrahydrofolate and methyl donors S-adenosyl-methionine are already in present bacteria enzymes and have been reported to be involved in benzene ring methylation [76]. As a result, methylation occurs and benzene is converted to o-xylene. Figure 20 shows a proposed biodegradation pathway for the biodegradation of methyl red by *P. aeruginosa*.

## 4. Conclusions

The aim of this study was to assess the biodegradation potential of *P. aeruginosa* for methyl red. The chosen bacterial strain degraded the textile azo dye methyl red more effectively. The optimal conditions determined were 3 days incubation time, 40 °C temperature, 20 ppm concentration of dye, pH 9, 1666.67 mg/L glucose concentration, 1000 mg/L urea concentration, 666.66 mg/L NaCl, and hydroquinone (redox mediator) at the concentration of 66.66 mg/L in a one-at-a-time factor optimization. The optimal conditions of pH, incubation time, temperature, and dye concentration influenced the maximum degradation of methyl red in a second-order quadratic model. The spectroscopic techniques FTIR, GCMS, and FTIR were used to characterize the metabolites formed under these optimal conditions. The spectroscopic data demonstrated that bacteria degraded the corresponding dye to benzoic acid and o-xylene by subsequent enzymatic actions. It was revealed that the significant metabolites were benzoic acid and o-xylene. The azo linkage of the methyl red dye was broken by enzyme azoreductase of the bacterial system, leading to the formation of 2-aminobenzoic acid ring and N,N-dimethylaminobenzene. Benzoic acid was formed by bacterial enzyme azo reductase reducing the azo linkage of methyl red dye, then followed by the deamination and then reduction of the 2-aminobenzoic acid ring. The other part of the dye molecule N,N-dimethylaminobenzene was converted to N,N-dimethyl benzene by deamination reaction. The deamination and later reductive demethylation of N,N-dimethyl benzene transformed it to aniline. Bacterial enzyme deaminase by deamination action changed the aniline molecule to benzene. After subsequent demethylation, benzene was converted to o-xylene. The effective degradation of methyl red was optimal using *P. aeruginosa*; therefore, it can be utilized as a suitable bacterial strain for the degradation of methyl red and other azo dyes. Some limitations are connected with biodegradation of dyes using *P. aeruginosa*, such as high concentration of dyes and lack of suitable physiochemical and environmental conditions. So, these conditions should be considered during the degradation study. Further research work is needed to isolate this bacterium from dye-contaminated sites and its degradation potential for other azo dyes should be reported. The enzymatic profiling and genetic sequencing of this bacterium should be determined using modern biotechnological approaches.

## Figures and Tables

**Figure 1 ijerph-19-09962-f001:**
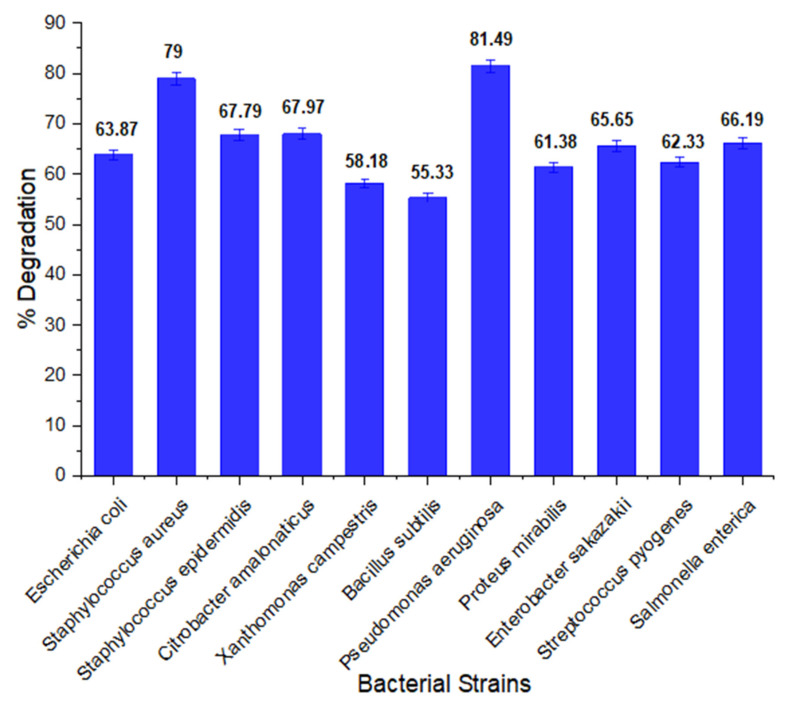
Percent degradation of methyl red by different bacteria.

**Figure 2 ijerph-19-09962-f002:**
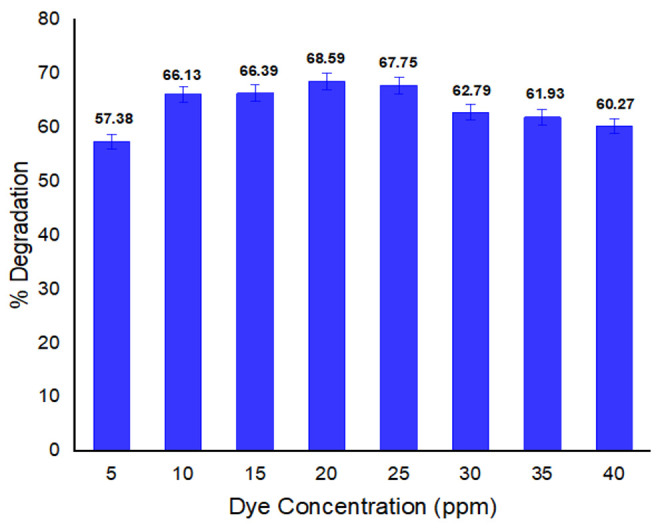
Dye concentration impact on % degradation.

**Figure 3 ijerph-19-09962-f003:**
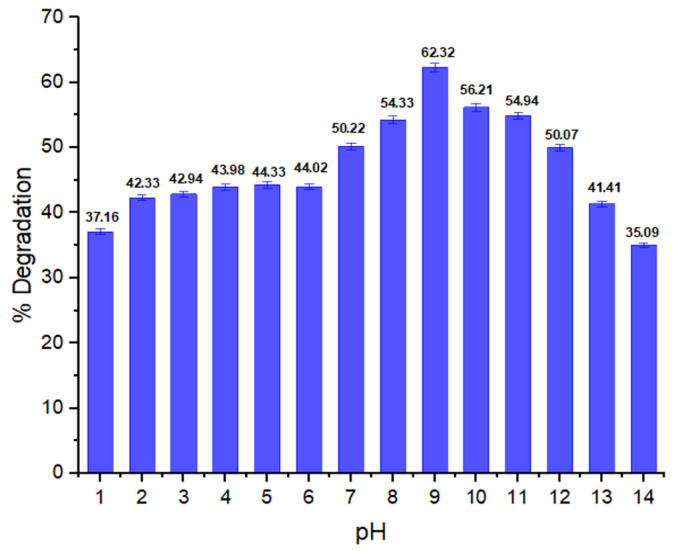
pH effect on % degradation of methyl red dye.

**Figure 4 ijerph-19-09962-f004:**
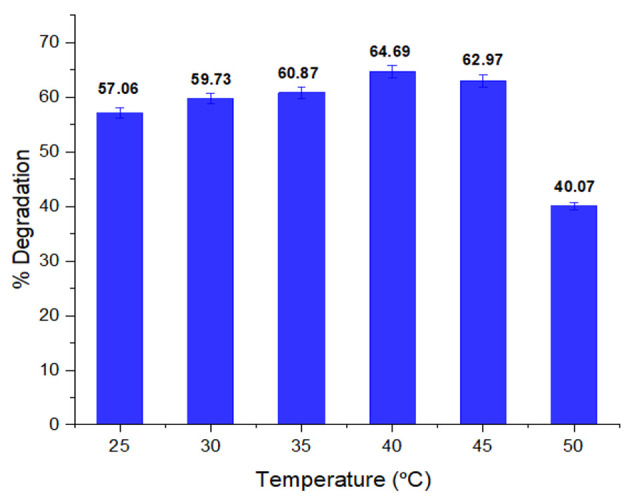
Temperature impact on % degradation of methyl red dye.

**Figure 5 ijerph-19-09962-f005:**
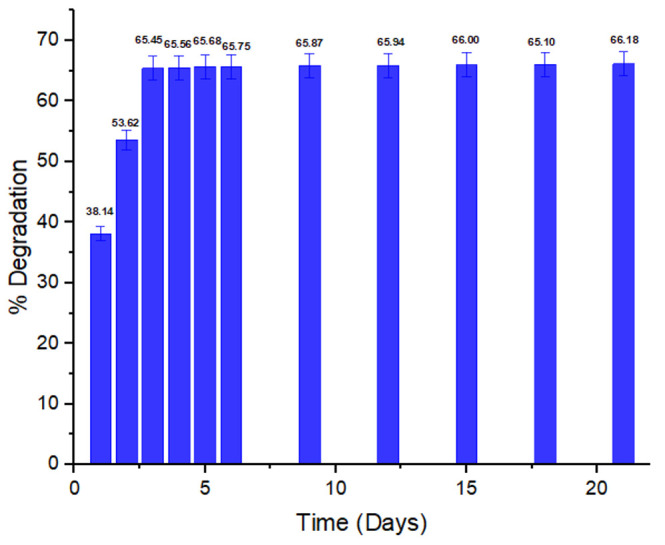
Incubation time (days) impact on % degradation.

**Figure 6 ijerph-19-09962-f006:**
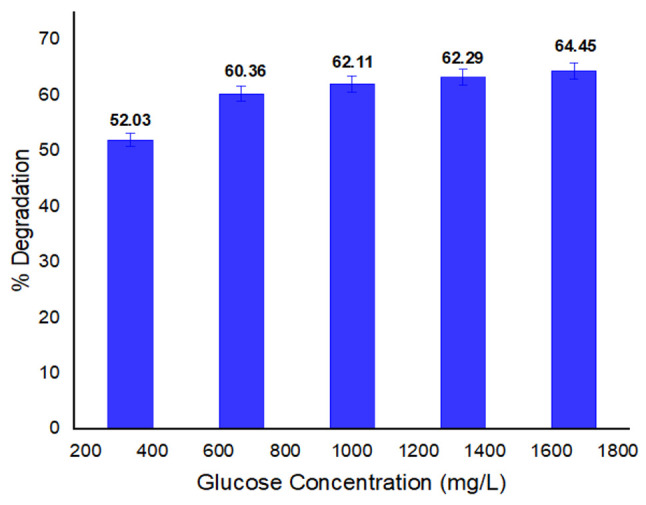
Glucose concentration and its impact on percent degradation of methyl red.

**Figure 7 ijerph-19-09962-f007:**
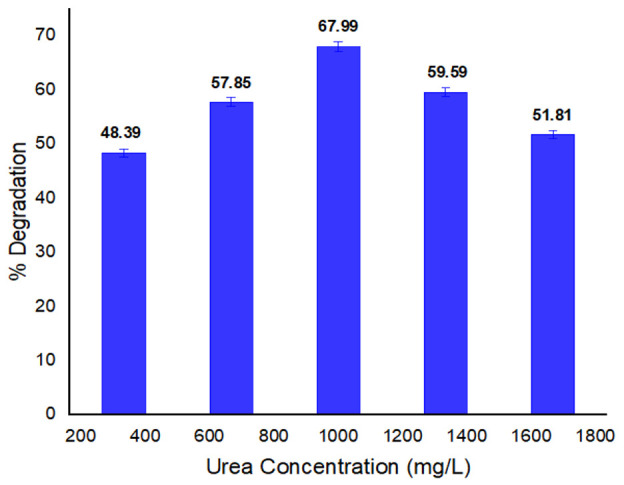
Urea concentration effect on methyl red % degradation.

**Figure 8 ijerph-19-09962-f008:**
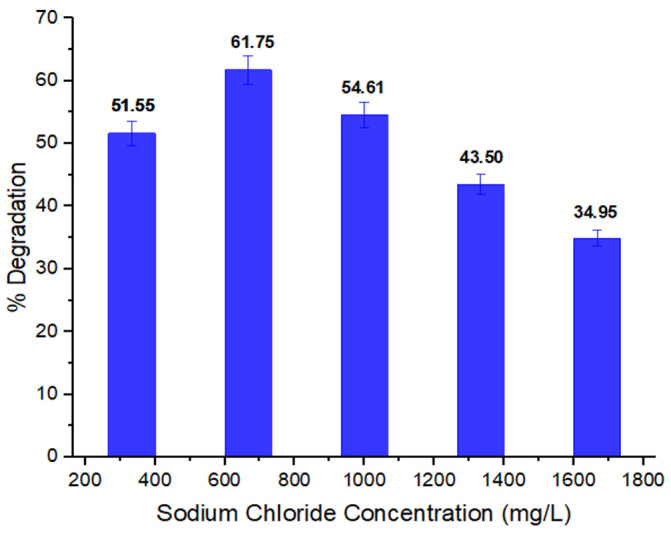
Effect of sodium chloride (mg/L) on dye % degradation.

**Figure 9 ijerph-19-09962-f009:**
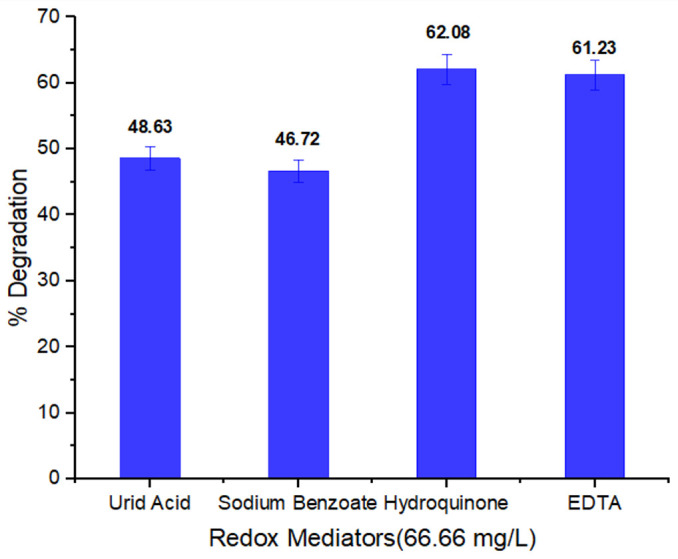
Redox mediators impact on (66.66 mg/L) on methyl red % degradation.

**Figure 10 ijerph-19-09962-f010:**
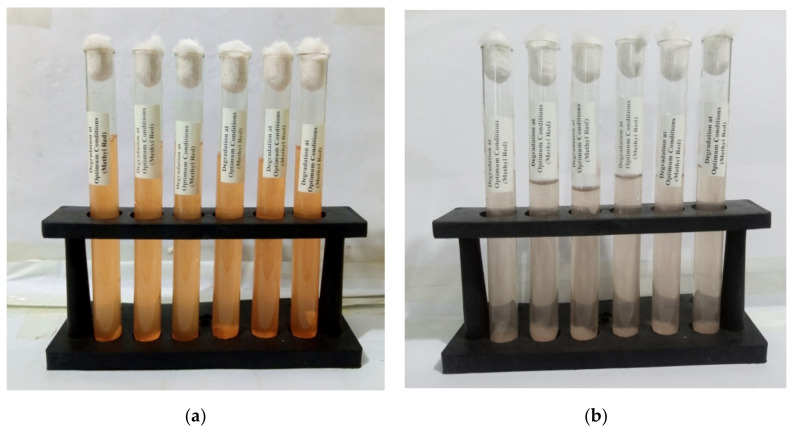
Methyl red dye (**a**) before treatment and (**b**) after treatment of *P. aeruginosa*.

**Figure 11 ijerph-19-09962-f011:**
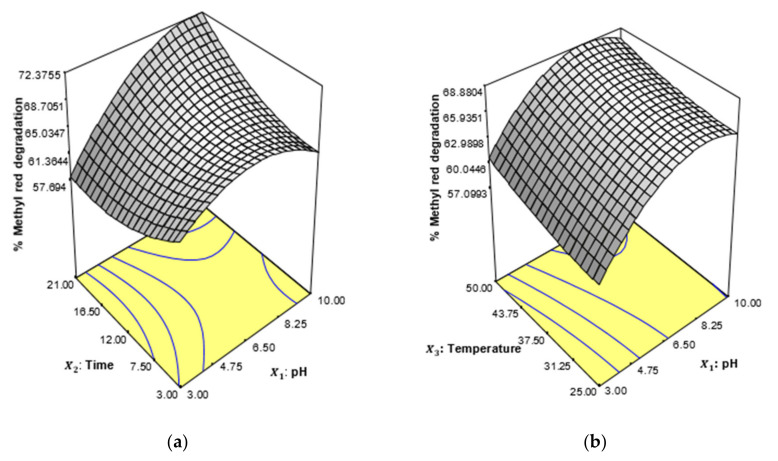
Response surface plot for the interaction of (**a**) pH and incubation time (**b**) pH and temperature (**c**) incubation time and dye concentration (**d**) temperature and dye concentration for the biodegradation of methyl red using *P. aeruginosa*.

**Figure 12 ijerph-19-09962-f012:**
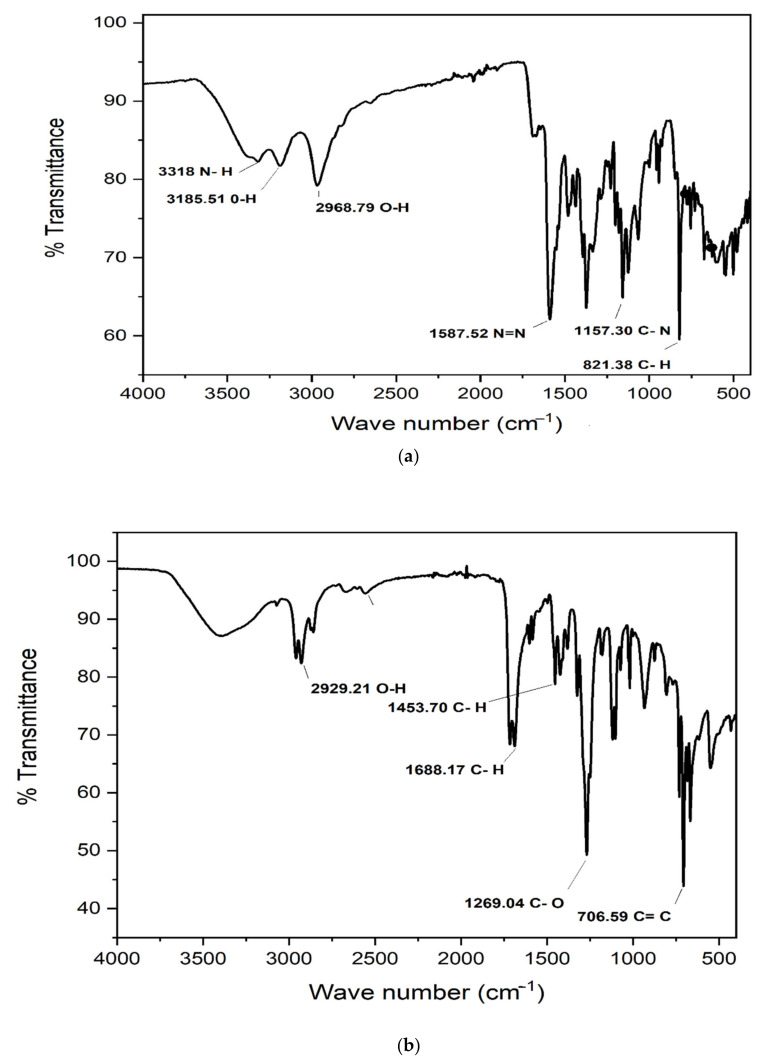
(**a**) FTIR spectra of methyl red before biodegradation (**b**) FTIR spectra of methyl red dye after *P. aeruginosa* degradation.

**Figure 13 ijerph-19-09962-f013:**
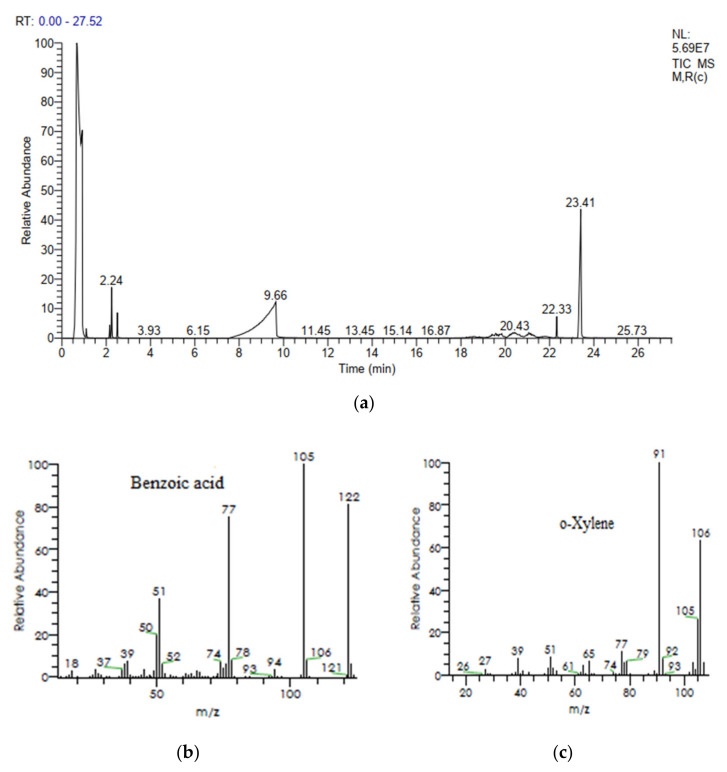
(**a**) GC chromatogram of methyl red after bacterial degradation; (**b**,**c**) GC–MS chromatograms of two important metabolites.

**Figure 14 ijerph-19-09962-f014:**
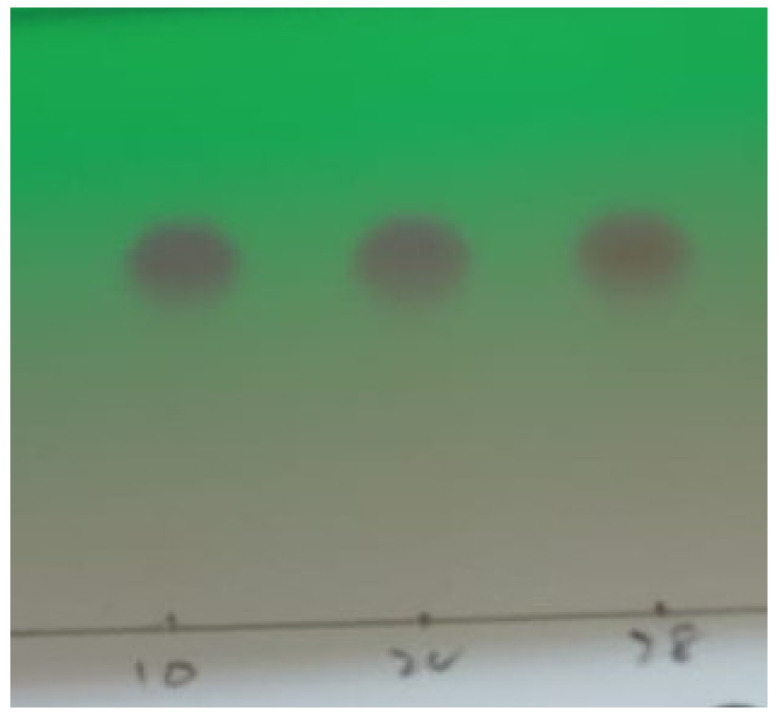
TLC profiling under UV light.

**Figure 15 ijerph-19-09962-f015:**
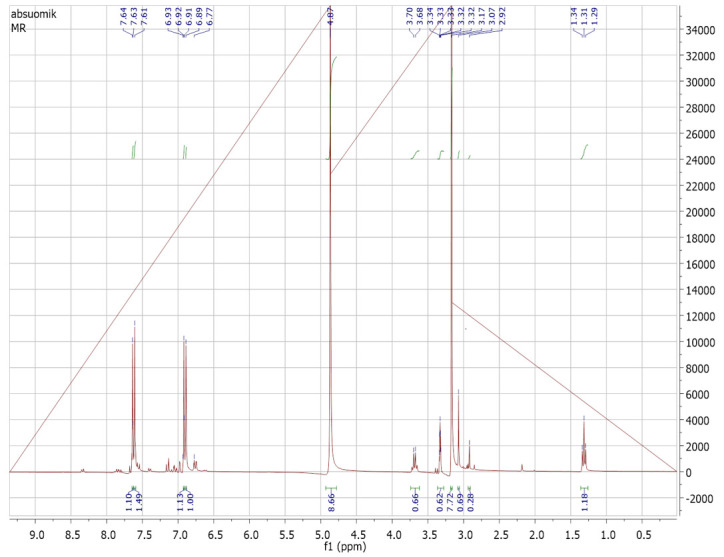
^1^H NMR of the methyl red.

**Figure 16 ijerph-19-09962-f016:**
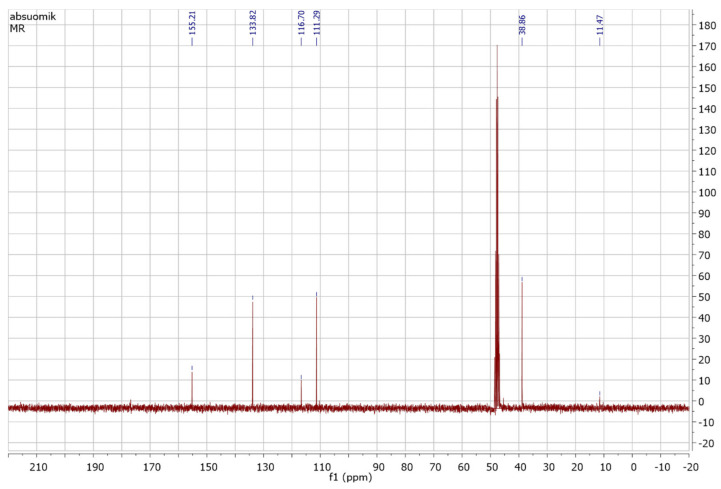
^13^C NMR of methyl red dye.

**Figure 17 ijerph-19-09962-f017:**
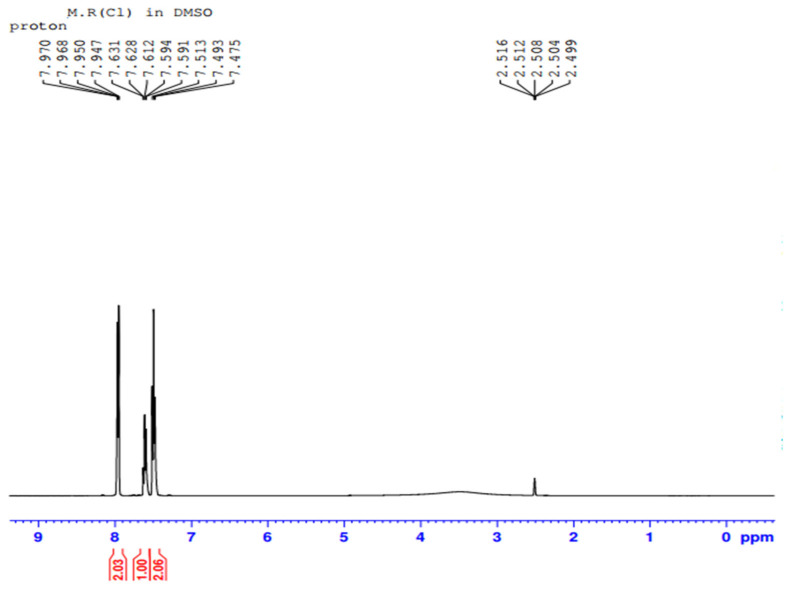
^1^H NMR of metabolite.

**Figure 18 ijerph-19-09962-f018:**
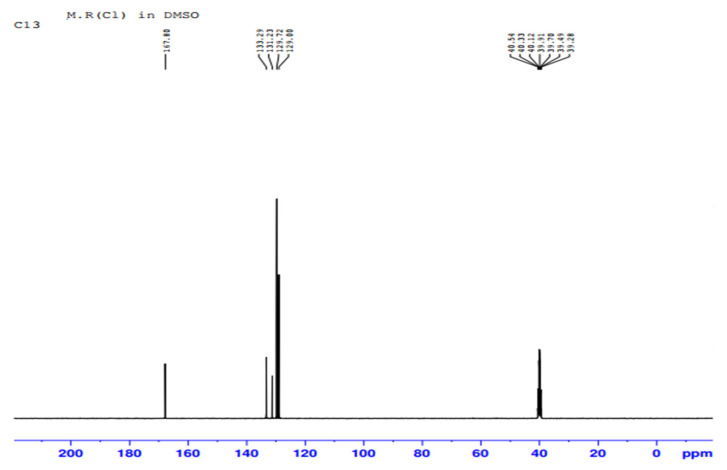
^13^C NMR of the metabolite.

**Figure 19 ijerph-19-09962-f019:**
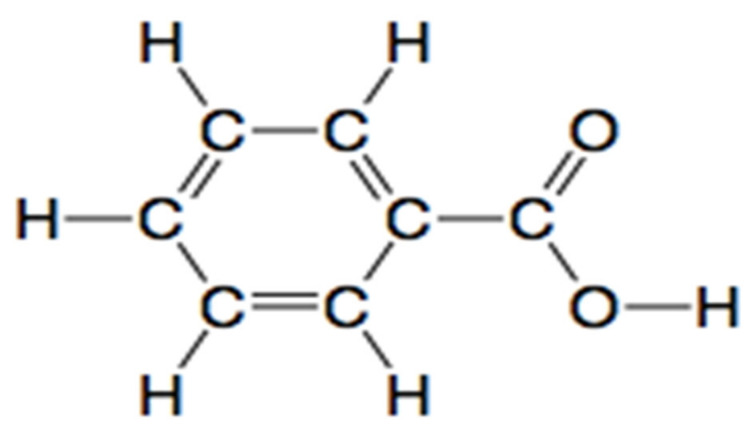
Chemical structure of an isolated metabolite (benzoic acid).

**Figure 20 ijerph-19-09962-f020:**
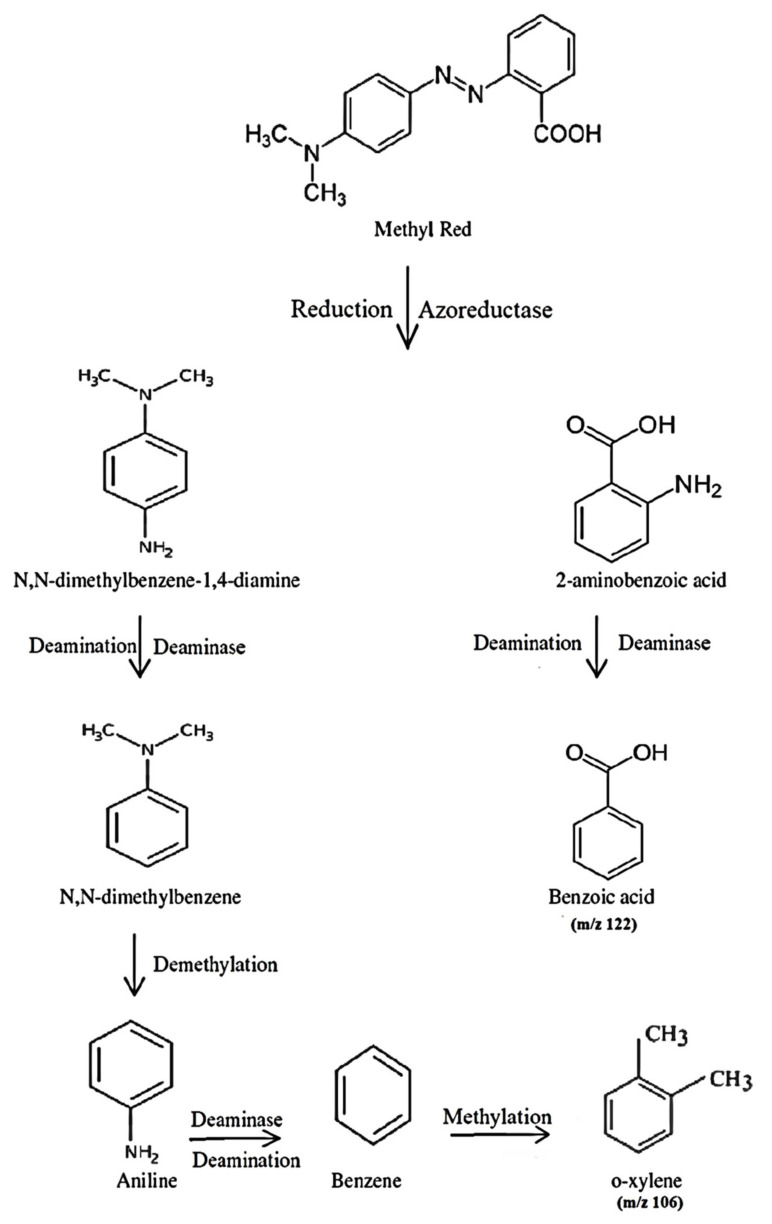
Proposed degradation mechanism of methyl red by *P. aeruginosa*.

**Table 1 ijerph-19-09962-t001:** The decolorization potential of different bacterial strains for dyes.

S.No Dyes	Bacterial Strain	%Decolorization	References
1. Reactive Red 180	*Citrobacter* sp.	90	[28]
2. Reactive Red 141	*Staphylococcus hominis*	93.24	[28]
3. Golden Yellow HER	*Brevibacillus laterosporus*	87	[29]
4. Disperse Orang SRL	*Acinetobacter* sp.	90.2	[30]
5.Victoria Blue R	*Acinetobacter calcoaceticus*	94.5	[30]
6. Direct Blue 71	*Rhizobium* sp.	95	[31]
7. Methyl Orange	*Pseudomonas aeruginosa*	88.23	[32]
8 Reactive Black 5	*Aeromonas hydrophila*	76	[33]
9. Direct Orange 16	*Micrococcus luteus*	96	[34]
10. Reactive Green 19A	*Micrococcus glutamicus*	96	[34]
11. Disperse Blue 284	*Klebsiella pneumoniae*	95	[35]
12. Red RBN	*Proteus mirabilis*	95	[36]
13. C.I. Reactive Red 195	*Enterococus faecalis*	99.5	[37]
14. Reactive Black 5	*Clostridium bifermentans*	90	[38]
15. Methyl red	*Zhihengliuella* sp.	98.87	[39]
16. Acid Blue 113	*Sphingomonas melonis*	80	[40]
17. Direct Black G	*Anoxybacillus* sp.	98.39	[40]
18. Acid Blue 113	*S.Boydii*	96	[40]

**Table 2 ijerph-19-09962-t002:** Analysis of central composite design for the degradation of methyl red.

Factors	Units	Code	Levels		
			−1	0	+1
pH	-	$X_{1}$	3	6.5	10
Dye concentration	ppm	$X_{2}$	5	22.5	40
Incubation time	day	$X_{3}$	3	21	12
Temperature	°C	$X_{4}$	25	37.5	50

**Table 3 ijerph-19-09962-t003:** Central composite design matrix.

Run	$X_{1}$	$X_{2}$	$X_{3}$	$X_{4}$	Observed Value	Predicted Value
1	−1	1	−1	1	47.61	36.95
2	1	1	1	−1	37.20	46.56
3	1	1	−1	−1	31.40	35.92
4	0	2	0	0	37.95	45.85
5	1	−1	−1	−1	56.60	57.80
6	0	0	0	0	58.20	56.23
7	0	0	2	0	57.89	57.80
8	0	0	0	0	64.20	62.29
9	−1	−1	−1	−1	58.43	65.92
10	1	−1	1	−1	75.40	72.90
11	2	0	0	0	68.30	65.69
12	0	0	0	−0.571	68.20	76.14
13	1	1	1	1	70.12	69.50
14	1	−1	1	1	72.90	70.59
15	−1	−1	−1	1	67.80	64.02
16	−1	−1	1	1	63.21	68.58
17	0	−2	0	0	42.30	44.82
18	0	0	0	2	61.80	56.29
19	0	0	0	0	72.15	73.08
20	0	0	0	0	81.20	77.27
21	0	0	0	0	71.23	63.90
22	−1	1	1	−1	69.20	73.54
23	0	0	−2	0	61.20	61.19
24	0	0	0	0	68.93	65.94
25	1	−1	−1	1	69.87	67.28
26	−2	0	0	0	69.20	67.28
27	1	1	−1	1	65.80	67.28
28	−1	1	−1	−1	69.73	67.28
29	−1	1	1	1	59.87	67.28
30	−1	−1	1	−1	69.20	67.28

**Table 4 ijerph-19-09962-t004:** ANOVA indicating percent degradation of methyl red dye.

Source	Sum of Squares	DF	Square Values	F-Value	*p*-Value
Model	3011.62	14	215.12	4.35	0.0038
$X_{1}$	328.93	1	328.93	6.65	0.0210
$X_{4}$	1546.70	1	1546.70	31.26	<0.0001
$X_{1}{}^{2}$	434.09	1	434.09	8.77	0.0097
$X_{4}{}^{2}$	375.73	1	375.73	7.59	0.0147
$X_{1}X_{2}$	181.51	1	181.51	3.67	0.0074
Lack of fit	-	-	-	-	0.1246
Cor Total	3753.90	29			
Mean	PRESS	Adeq Precision	R-Squared	Adj R-Squared	Std. Dev.
62.20	4180.75	7.931	0.8023	0.6177	7.03

**Table 5 ijerph-19-09962-t005:** Identified metabolites in the dye-degraded mixture from GCMS results.

S.No	Metabolite	Peak Area	Retention Time	Chemical Formula	Molecular Weight
1.	Benzoic acid	13.10	9.58	C_7_H_6_O_2_	122
2.	o-Xylene	2.45	2.24	C_8_H_10_	106

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
