# Peer review of "Biodegradation of Azo Dye Methyl Red by Pseudomonas aeruginosa: Optimization of Process Conditions"

_ijerph, 2022, doi:10.3390/ijerph19169962_

Round 1

Reviewer 1 Report

1- Abstract needs to modify and It is better to write it more concisely.

2- Abbreviations must be complete in the first instance. (In Line 37: NMR)

3- Figure 22 have not mentioned in the text.

4- In the conclusion section, limitations and recommendations of this research should be highlighted.

5- In Lines 442, 543 & 578: please replace “shows” with “show”.

6- In Lines 566 & 571: Please replace “is shown” with “are shown”.

Author Response

Reviewer 1 Comment: Abstract needs to modify and It is better to write it more concisely. Response: The abstract has been revised and shortened accordingly. Comment: 2- Abbreviations must be complete in the first instance. (In Line 37: NMR) Response: The abbreviation has been completed accordingly. Comment: 3- Figure 22 have not mentioned in the text. Response: Mistake. Figure 22 has been mentioned in text. Thank you Comment: 4- In the conclusion section, limitations and recommendations of this research should be highlighted. Response: The conclusion has been revised to include limitations and recommendations. Comment: 5- In Lines 442, 543 & 578: please replace “shows” with “show”. Response: These mistakes have been corrected. Comment: 6- In Lines 566 & 571: Please replace “is shown” with “are shown”. Response: This mistake has been corrected. Thank you

Reviewer 2 Report

The manuscript presents an essential area of research on treatment of Azo dye Methyl Red using Pseudomonas aeruginosa. The manuscript is relevant to the scope of the Journal. This manuscript is very well written, the authors have performed systematic studies, and the data is sufficient to support some interesting results. Therefore, I recommend the publication of the manuscript after revision. Importantly, the authors need to clearly review their work and correct all the ambiguous and unclear sentences with some grammatical errors in the manuscript. Following are the detailed/ specific comments:

Abstract section. This section seems fine to me.

Introduction section

General comment on this section: The section seems fine to me. However, throughout this section there are such ambiguous and unclear sentences with some grammatical errors. Thus, significant revision is required

1.       In line 62: Please revise “colourings” to “colouring”

2.       In line 95: Please include “into” between “…results” and “additional…”

3.       In line 124: Please include “the” between “of” and “present study” .i.e. the aim of the present study....

Materials and Methods section: The section seems fine to me. However, throughout this section there are still such ambiguous and unclear sentences with some grammatical errors. Thus, significant revision is required.

Secondly, the authors should consider including a section on statistical analysis to clearly describe the statistical tests used for analysis or to test the hypothesis investigated in this study.

1.       In line 136: Please, put a full stop at the end of sentence 135.

2.       Lines 156-158 “....... Please, review this sentence. It’s not clear to me.

3.       In line 159: This sentence needs to be revised as it’s not clear – maybe the authors wanted to mean “....... transferred “to an” autoclave” …

4.       Line 160: This sentence is also not clear. Maybe the authors wanted to mean that "then, to inoculate bacteria in the medium, nutrient broth solution and test tubes were removed from autoclave and stored in the laminar flow hood" - without the word "machine". Please clarify.

5.       Lines 166-167: Revise this sentence to "5ml of dye solution from methyl red solution was added to each test tube" OR “each test tube was filled-up with .....”

6.       In Section 2.7, lines 180-182: Please, provide details on how these parameters were tested. You may include the details in the supplementary information.

7.       In line 183: please, provide a comma immediately after "In each case,"

8.       In line 189: Please, revise this from "has" to "had"

9.       In line 211: Please include “an” … i.e., in "an" incubator.

10.   In line 222: Please, provide a comma between “interval” and “the degraded…”

11.   Line 235: This sentence is not clear to me. Maybe the authors wanted to mean that “……. it’s a major salt that increase sea water salinity”.

12.   In line 241: Please add “an” … i.e., in "an" incubator.

13.   In line 248: Please add “a” … i.e., serves "a" nitrogen source….

14.   In line 251: Please delete one of the “of”

15.   Lines 256-258 “....... Please, review this sentence. It’s not clear to me.

16.   In line 258: Please add “use” … i.e., were also “used” a reference….

17.   Lines 285-287 “....... Please, review this sentence. It’s not clear to me. Maybe delete “was” between “extract” and “subjected…”.

18.   Line 291 “....... Please, rephrase this sentence. It’s not clear to me.

19.   Lines 306-307 “....... Please, rephrase this sentence. It’s also not clear to me.

Results and Discussion Section: The section seems fine to me. However, as other sections, also throughout this section there are such ambiguous and unclear sentences with some grammatical errors. Thus, significant revision is required.

1.       In line 317: Please, provide a comma immediately after "strains"

2.       In line 317 - 325: The authors state that “The degradation potential of different bacteria is different” however, its unclear on how they arrived at such conclusion/ statement. What statistical test did you use to determine that the degradation potential of the different bacteria is different, and at what confidence interval? Was really the biodegradation capacity of Pseudomonas aeruginosa - statistically different from Staphylococcus aureus (based on the results presented in Figure 2)? This goes back to my comment in the methods section about the data analysis and statistical tests used.

3.       lines 317 - 325: Please, un-bold the text.

4.       Based on the results presented in Figure 3, I doubt if there is really a statistically significant difference between the % degradation of the concentrations 10, 15,20 and 25 (ppm). The authors should conduct some statistical tests to establish whether there is a statistically significant difference between these concentrations, before they make such a big claim.

5.       In line 363: Please add “be” … i.e., may probably "be" due to …..

6.       In line 371: Please delete “was” … The percentage degradation “was” decreased…

7.       In line 376: Please, provide a comma immediately after "After three days,…."

8.       Lines 392-393: Revise this statement to “The breakdown activity reduced as the content of urea increased due to induced toxicity”. However, the authors need to conduct statistical tests to establish if there is a significant correlation between these parameters and the Dye biodegradation - clearly reporting the R2 and P values.

9.       In line 396: Please delete “was” … percentage breakdown activity “was” reduced ….

10.   In line 403: Please delete “this”…

11.   In line 403: Please, delete a comma immediately after "was found"

12.   In line 409: Please reviseE.Coli” to “E. coli

13.   Line 415 “....... Please, review this sentence. It’s not clear to me. Maybe add “and” after the comma.

14.   Lines 419-420 “....... Please, provide the citation of the recent study you're referring to.

15.   Lines 418 – 422 …. Is the performance of hydroquinone really statistically different from EDTA? According to the results presented in Figure 10 - these redox mediators have a similar performance.

16.   In line 432: Please delete one of “the”…

17.   In line 432: Why is "C" capital. Please, revise accordingly

18.   In lines 434-435: Please, provide a comma immediately after "time", and after “Temperature”

19.   In line 445: Please, revise this from "has" to "had"

20.   In line 466: Please, include a section in the methods section for the statistical tests used in your study and their confidence intervals.

21.   In line 474: This is plural - change “has” to “have” significant

22.   Line 480 -485 “....... Please, rephrase these sentences. They’re not clear to me.

23.   In line 498: Please, move Table 2 to the supplementary information

24.   In line 531-533: Please, revise this sentence. Its also not clear to me.

25.   Line 544 – 545: Please discuss your results.

26.   Line 555 – 558: Please discuss your results.

27.   In line 584: Please delete “is”…

28.   In line 599: Please delete “have been”…

Conclusion Section: This section seems fine to me.

1.       In line 633: Please delete “has been”… and replace it with “was”

2.       In line 635: Please delete “has”…

3.       In line 636: Please delete “has”…

4.       In line 637: Please delete “has been”… and replace it with “was”

Author Response

Reviewer 2

Comment: The manuscript presents an essential area of research on treatment of Azo dye Methyl Red using Pseudomonas aeruginosa. The manuscript is relevant to the scope of the Journal. This manuscript is very well written, the authors have performed systematic studies, and the data is sufficient to support some interesting results. Therefore, I recommend the publication of the manuscript after revision.

Response: Thank you referee for your esteemed remarks and appreciation of our work.

Importantly, the authors need to clearly review their work and correct all the ambiguous and unclear sentences with some grammatical errors in the manuscript Following are the detailed/ specific comments:

Comment: Abstract section. This section seems fine to me.

Response: Thank you referee for your appreciation.

Comment: Introduction section. General comment on this section: The section seems fine to me. However, throughout this section there are such ambiguous and unclear sentences with some grammatical errors. Thus, significant revision is required.

Response: The manuscript text has been revised.

       Comment: In line 62: Please revise “colourings” to “colouring”

       Response: This mistake has been corrected. Thank you.

       Comment: In line 95: Please include “into” between “…results” and “additional…”

       Response: This word has been introduced accordingly

       Comment: In line 124: Please include “the” between “of” and “present study” .i.e. the aim of the present study....

        Response: These words have been introduced accordingly.

Comment: Materials and Methods section: The section seems fine to me. However, throughout this section there are still such ambiguous and unclear sentences with some grammatical errors. Thus, significant revision is required.

Response: The manuscript has been revised and errors have been removed.

Comment: Secondly, the authors should consider including a section on statistical analysis to clearly describe the statistical tests used for analysis or to test the hypothesis investigated in this study.

Response: The data analysis has been mentioned in the Materials and Method section.

     Comment: In line 136: Please, put a full stop at the end of sentence 135.

     Response: This mistake has been corrected.

     Comment: Lines 156-158 “....... Please, review this sentence. It’s not clear to me.

      Response: This sentence has been reviewed accordingly.

  Comment: In line 159: This sentence needs to be revised as it’s not clear – maybe the authors wanted to mean “....... transferred “to an” autoclave” …

  Response: This mistake has been corrected. Thank you

  Comment: Line 160: This sentence is also not clear. Maybe the authors wanted to mean that "then, to inoculate bacteria in the medium, nutrient broth solution and test tubes were removed from autoclave and stored in the laminar flow hood" - without the word "machine". Please clarify.

  Response: Yes, the authors wanted to mean "then, to inoculate bacteria in the medium, nutrient broth solution and test tubes were removed from autoclave machine and stored in the laminar flow hood.

Comment: Lines 166-167: Revise this sentence to "5ml of dye solution from methyl red solution was added to each test tube" OR “each test tube was filled-up with .....”

Response: The sentence has been corrected accordingly. Thank you

 Comment: In Section 2.7, lines 180-182: Please, provide details on how these parameters were tested. You may include the details in the supplementary information.

Response: The percentage degradation was calculated in each case. The effect of various physiochemical conditions or parameters like dye concentration, pH, time, temperature, glucose concentration, NaCl concentration, Urea concentration and redox mediators. were testes in separate experiments.  The optimum conditions found were 20ppm Methyl red concentration, pH 9, temperature 40°C, 1666.67mg/L glucose concentration, 666.66mg/L sodium chloride concentration, 1000mg/L urea concentration, 3 days incubation duration, and 66.66mg/L hydroquinone. The other experimental parameters/ conditions were same as those described earlier. Consequently, the percentage degradation reached to 88.37 when these optimal physiochemical conditions were employed in a single experiment.  

Comment: In line 183: please, provide a comma immediately after "In each case,"

Response: This mistake has been corrected. Thank you

Comment: In line 189: Please, revise this from "has" to "had"

Response: This mistake has been corrected accordingly.

Comment: In line 211: Please include “an” … i.e., in "an" incubator.

Response: This mistake has been corrected.

Comment: In line 222: Please, provide a comma between “interval” and “the degraded…”

Response: This mistake has been corrected.

Comment: Line 235: This sentence is not clear to me. Maybe the authors wanted to mean that “……. it’s a major salt that increase sea water salinity”.

Response: Yes the authors means ; it’s a major salt that increase sea water salinity”.

Comment: In line 241: Please add “an” … i.e., in "an" incubator.

Response: This mistake has been corrected.

Comment: In line 248: Please add “a” … i.e., serves "a" nitrogen source….

Response: This mistake has been corrected.

Comment: In line 251: Please delete one of the “of”

Response: This mistake has been corrected.

Comment:   Lines 256-258 “....... Please, review this sentence. It’s not clear to me.

Response: These sentences has been revised.

Comment: In line 258: Please add “use” … i.e., were also “used” a reference….

Response: The sentence has been revised.

Comment: Lines 285-287 “....... Please, review this sentence. It’s not clear to me. Maybe delete “was” between “extract” and “subjected…”.

Response: The sentence has been revised.

Comment: Line 291 “....... Please, rephrase this sentence. It’s not clear to me.

Response: The sentence has been revised.

Comment: Lines 306-307 “....... Please, rephrase this sentence. It’s also not clear to me.

Response: This sentence has been revised.

Comment: Results and Discussion Section: The section seems fine to me. However, as other sections, also throughout this section there are such ambiguous and unclear sentences with some grammatical errors. Thus, significant revision is required.

Response: This section has been revised accordingly.

Comment: In line 317: Please, provide a comma immediately after "strains"

Response: This mistake has been corrected and comma has been added.

Comment: In line 317 - 325: The authors state that “The degradation potential of different bacteria is different” however, its unclear on how they arrived at such conclusion/ statement. What statistical test did you use to determine that the degradation potential of the different bacteria is different, and at what confidence interval?

 Response: Thank you referee. The potential of the investigated bacteria varies on the basis of their degradation capacity of methyl red dye. The percent degradation potential of the respective dyes were evaluated according to equation 1. For every test sample, analysis was conducted in triplicate and the result was recorded in average as indicated in Figure 1 in the revised manuscript.

Comment: Was really the biodegradation capacity of Pseudomonas aeruginosa - statistically different from Staphylococcus aureus (based on the results presented in Figure 2)?

 Response: There is no much significant difference in biodegradation capacity of Pseudomonas aeruginosa compared to Staphylococcus aureus although it appeared that Pseudomonas aeruginosa has better degradation capacity due to azoreductase enzyme which breaks faster the azo bond of the dye molecule, this enzyme is most commonly found in this bacteria.

Comment: This goes back to my comment in the methods section about the data analysis and statistical tests used.

Response: All the experiments including control groups throgh out the experiments were repeated in triplicates, and the results obtained were presented as the Mean ± Standard deviation. The effect of the process conditions and the interaction for the evaluation of optimum degrdation efficiency of bacteria was analyzed according to ANOVA at P<0.05. This is in Table 3. 

Comment: lines 317 - 325: Please, un-bold the text.

Response: The text has been un-bold accordingly.

Comment: Based on the results presented in Figure 3, I doubt if there is really a statistically significant difference between the % degradation of the concentrations 10, 15, 20 and 25 (ppm). The authors should conduct some statistical tests to establish whether there is a statistically significant difference between these concentrations, before they make such a big claim.

Response: Thank you referee. The effect of dye concentration was optimized based on the optimum bacteria degradation efficiency in Figure 1. There is slight significance difference in the degradation efficiency of the bacteria base on dye concentration. The effect of the optimum conditions was evaluated using central composite design (CCD). The statistical significance of the effect of the interaction of the process condition was analysed using the ANOVA of the response surface methodology at P<0.05.

Comment: In line 363: Please add “be” … i.e., may probably "be" due to …..

Response: This mistake has been corrected in text.

Comment: In line 371: Please delete “was” … The percentage degradation “was” decreased…

Response: This mistake has been corrected in manuscript.

Comment: In line 376: Please, provide a comma immediately after "After three days,…."

Comment: Lines 392-393: Revise this statement to “The breakdown activity reduced as the content of urea increased due to induced toxicity”.

Response: The statement has been revised accordingly.

Comment: However, the authors need to conduct statistical tests to establish if there is a significant correlation between these parameters and the Dye biodegradation - clearly reporting the R2 and P values.

Response: Statistical analysis was conducted on four significant parameters which are pH, dye concentration, incubation time and temperature to investigate their effect on the biodegradation efficiency of the bacteria. The result of the R2 and P-Values based on the linear, quadratic and effect of the interaction of the process variables is presented in Table 3.

Comment: In line 396: Please delete “was” … percentage breakdown activity “was” reduced ….

Response: This mistake has been corrected in manuscript.Thank you

Comment:   In line 403: Please delete “this”…

Response: This mistake has been corrected in manuscript. Thank you

Comment: In line 403: Please, delete a comma immediately after "was found"

Response: This mistake has been corrected in manuscript. Thank you

Comment: In line 409: Please revise “E.Coli” to “E. coli

Response: This mistake has been corrected in manuscript.

Comment: Line 415 “....... Please, review this sentence. It’s not clear to me. Maybe add “and” after the comma.

Response: Response: This sentence has been revised accordingly.

Comment: Lines 419-420 “....... Please, provide the citation of the recent study you're referring to.

Response: The citation has been added and mentioned in the manuscript text.

Comment: Lines 418 – 422 …. Is the performance of hydroquinone really statistically different from EDTA? According to the results presented in Figure 10 - these redox mediators have a similar performance.

Response: Different redox mediators have different performances. There is no much significant difference in our experiment.

Comment: In line 432: Please delete one of “the”…

Response: This mistake has been corrected in manuscript.

Comment: In line 432: Why is "C" capital. Please, revise accordingly

Response: This mistake has been corrected in manuscript.

Comment: In lines 434-435: Please, provide a comma immediately after "time", and after “Temperature”

Response: This mistake has been corrected in manuscript.Thank you

Comment: In line 445: Please, revise this from "has" to "had"

Response: This mistake has been corrected in manuscript.

Comment: In line 466: Please, include a section in the methods section for the statistical tests used in your study and their confidence intervals.

Response: This section has been included accordingly. Thank you

Comment: In line 474: This is plural - change “has” to “have” significant

Response: This mistake has been corrected in manuscript.

Comment: Line 480 -485 “....... Please, rephrase these sentences. They’re not clear to me.

Response: These sentences have been revised.

Comment: In line 498: Please, move Table 2 to the supplementary information

Response: This mistake has been corrected in manuscript.

Comment: In line 531-533: Please, revise this sentence. Its also not clear to me.

Response: These sentences has been revised accordingly.

Comment: Line 544 – 545: Please discuss your results.

Response: The results has been discussed.

Comment: Line 555 – 558: Please discuss your results.

Response: The results has been discussed.

Comment: In line 584: Please delete “is”…

Response: This mistake has been corrected in manuscript.

Comment: In line 599: Please delete “have been”… 

Response: This mistake has been corrected in manuscript.

Comment: Conclusion Section: This section seems fine to me.

Response: Thank you very much for your appreciation.

Comment: In line 633: Please delete “has been”… and replace it with “was”

Response: This mistake has been corrected in manuscript.

Comment: In line 635: Please delete “has”…

Response: This mistake has been corrected in manuscript.

Comment: In line 636: Please delete “has”…

Response: This mistake has been corrected in manuscript.

Comment: In line 637: Please delete “has been”… and replace it with “was”

Response: This mistake has been corrected in manuscript.

Reviewer 3 Report

There are a lot of studies on the applications of Pseudomonas aeruginosa in dye biodegradation, including the biodegradation of some azo dyes (Zhang, Q., Xie, X., Xu, D., Hong, R., Wu, J., Zeng, X., ... & Liu, J. (2021). Accelerated azo dye biodegradation and detoxification by Pseudomonas aeruginosa DDMZ1-2 via fructose co-metabolism. Environmental Technology & Innovation, 24, 101878; Yogananth, N., Vijaya, P. P., Aishwaryalakshmi, R., & Ali, M. S. (2012). Isolation, Purification and Characterization of Oxygen Insensitive Azoreductase from Pseudomonas aeruginosa and Biodegradation of Azo Dye-Methyl Red. Journal of Advanced Laboratory Research in Biology, 3(4), 285-289). However, the authors clearly explained the degradation mechanism of Methyl red by Pseudomonas aeruginosa, which is very interesting. In addition, the degradation efficiency was also improved significantly. Therefore, this manuscript could be accepted after some modifications.

(1)   Are there any reports on the degradation mechanisms of other azo dyes by Pseudomonas aeruginosa? The corresponding literature should be cited.

(2)   Some figures such as Fig. 11 and Fig. 12 (Fig. 14 a and 14b, Fig. 15 and Fig. 16) could be drawn in one image

Author Response

Reviewer 3

There are a lot of studies on the applications of Pseudomonas aeruginosa in dye biodegradation, including the biodegradation of some azo dyes (Zhang, Q., Xie, X., Xu, D., Hong, R., Wu, J., Zeng, X., ... & Liu, J. (2021). Accelerated azo dye biodegradation and detoxification by Pseudomonas aeruginosa DDMZ1-2 via fructose co-metabolism. Environmental Technology & Innovation, 24, 101878; Yogananth, N., Vijaya, P. P., Aishwaryalakshmi, R., & Ali, M. S. (2012). Isolation, Purification and Characterization of Oxygen Insensitive Azoreductase from Pseudomonas aeruginosa and Biodegradation of Azo Dye-Methyl Red. Journal of Advanced Laboratory Research in Biology, 3(4), 285-289).

Comment: However, the authors clearly explained the degradation mechanism of Methyl red by Pseudomonas aeruginosa, which is very interesting. In addition, the degradation efficiency was also improved significantly. Therefore, this manuscript could be accepted after some modifications.

Response: Thank you referee for your esteemed remarks and appreciation of our work.

Comment: (1) Are there any reports on the degradation mechanisms of other azo dyes by Pseudomonas aeruginosa? The corresponding literature should be cited.

Response: The latest research work and degradation mechanisms of other azo dyes by Pseudomonas aeruginosa has been cited in manuscript.

In literature there are reports on the degradation mechanism of other azo dyes brought about by P. aeruginosa. Recently, Ullah et al has reported an insight into the mechanism of biodegradation of other azo dyes Methyl orange and Brown 706 by P. aeruginosa [30,46].

     Comment: Some figures such as Fig. 11 and Fig. 12 (Fig. 14 a and 14b, Fig. 15 and Fig. 16) could be drawn in one image.

      Response:Thank you, this has been done in Fig. 11 and Fig. 12, Fig.15 and Fig 16 were shown in one figure. Sorry; Fig. 14 a and 14b has better resolution in isolated form. 

Reviewer 4 Report

Line 23: not sure why 11 is typed in brackets after the word eleven. This can be removed

Line 25: P. aeruginosa can be used the second (and following) time. No need to write Pseudomonas in full after the first time. Check throughout the rest of the text.

The first sentence of the introduction (line 50 to 52) is more or less the same as the second sentence (line 52-54). Better to merge both sentences into 1 sentence. Keep both references. 

Line 58: without treatment to THE environment. The needs to be added

Line 63 -  65: check sentence, its the same as the previous sentence and the sentence itself makes no sense. Other life forms in the water vs other living organisms?

Line 68: remove the word induce. 

Line 68-70: this sentence is yet again a repetition of previous sentences regarding the disturbance on the aquatic life. Merge with previous sentences and keep references

Line 72: put a comma before particularly and a comma after fish.   

Line 73-74: and also affect ... not affecting. Check sentence in general. Exhibit should be replaced with cause

Line 81-84: check sentence and rewrite. Sentence is too long and gibberish near the end.

Line 84-86: check sentence, incorrect. Alternative: Discharge of dyes without any proper treatment can influence aquatic life by blocking light penetration.

Line 89: wastewater not waste water

Line 93: limitations 

Line 95: that results IN additional...

Line 100: remove "over the time"

Line 106-107: literature overview.... yet only 1 reference is given? Are there not more review papers discussing this more in detail?

Line 107-108: the bacterial body systems? This is not proper English or a proper way to refer to bacteria.

Line 109: better reference(s) need to be added, reference 25 is not good enough.

Line 114: studies and not study

Line 114-115: not sure what the added value is of adding this extra sentence here about the Ikram et al study? E. coli is mentioned with the other bacteria above. 

Line 109-115: can be put in a table to make it easier to read.

Line 115: bacterial system.. ? This is not correct English.

Line 123: Sentence starts wrong. Better to write e.g.: However, to date only limited studies are available in the literature...  Also: no reference is given here about this statement? 

Line 124 and further: P. auruginosa, no need to always write Pseudomonas

Line 124-126: re-read sentence and change, this is a badly written sentence

Line 126: impcts? Impacts? 

Line 131: metabolites

Line 133-137: brand of the chemicals need to be added

Line 142-145: names need to in italic. Also improve sentence by e.g.: the biodegradation efficiency of the following bacteria was tested: .... Also: what are the specific strains? Why are the deposit numbers of the databases not mentioned (where are the strains from?). Some of the bacteria mentioned can be human pathogens. Now it is only mentioned they are from the Abdul Wali Khan University Mardan in Pakistan? But more information should be provided. 

Also: what is the optimal growth temperatures for these bacterial strains? It is mentioned that growth tests were performed at 37°C but is this the best suited temperature for all the strains?

Line 150: 'about' 0,04g? what is about 0.04?? This is not very scientific. 

Line 157: what type of nutrient broth? Brand of the broth needs to be mentioned

Line 156-157: re-read sentence, adjust, not proper English

Line 158: glassware not glass wares

Line 158: were transferred autoclave? Not proper English. Sentence has to be rewritten. 

Line 158: sterilization at 121°C for 3 hours? Why 3 hours?? 

Line 161: not sure what machine stored means here?

Line 166: re-read sentence, something is wrong. E.g. to each tube 5ml dye was added ... and not what is written now.

Line 176: The bacterial strain with the highest percentage...

Line 183: by conducting three sets of experiments? I assume you mean the experiments were performed in triplicate?

Line 196: Xi is mentioned again?

Line 193: unsure why there are two summations shown in the formula at the last summation part

Line 200: not entirely clear what the authors mean with static conditions? Can they explain?

Line 202: space after each comma

Line 202-203: I assume the authors mean a control was also made for each concentration? A control without bacteria?

205: brand of the centrifuge? Also rewrite sentence, not proper English

208-214: there is no mentioning of the pH values tested? At what wavelengths was the measurement done? Same as mentioned above? 430nm? If so: is this correct? The absorption spectra of methyl red also depends on the pH of the solution.

Line 201: fourteen is ok, no need to add (14)

Line 223-224: samples are now filtered? In the previous paragraphs there is no mentioning of filtering?

Line 217: to investigate the effect of temperature effect? This is not proper English

Line 221: mention the tested temperatures in ascending order: 25, 30, 35, 40, 45 and 50°C. 

Line 221: are the tested temperatures logical? Will P. aeruginosa survive temperatures of 40 to 50°C ? Why was 37°C not included? Were all the other tests (conditions) performed at 37°C?

Line 227: not entirely sure what is ment here. The authors mean that the main carbon source was glucose? 

Line 228: different concentrations of glucose were added? But the glucose is stated as a mass not a concentration? What is the final concentration of the samples?

Line 235-237: not entirely sure what the authors mean here. They mean that the optimal osmotic pressure is important for the bacteria to survive? Or is the salinity also a factor that influences the dye itself? 

Line 241-246: unsure what is mentioned here. Do the authors mean that the growth of the bacteria was followed for 21 days and every 3 days a sample was taken and measured ? If so: how is this possible of the start volume is 15mL (10mL broth, line 161 and 5mL of dye, line 166) and each sample that is taken is 5mL (line 170)

Line 248-253: why was urea added? The bacteria were grown in nutrient (bacterial) broth, which provides all the necessary nutrients? 

Line 256-257: incorrect English, needs to be rephrased.

Line 258: I assume the 1mg/15mL is the final concentration used? So 1mg of each tested substance was added?

Line 273: was crushed, unsure what the authors mean with crushed

Line 274: why centrifuged for 30 minutes here compared to 10 minutes in previous paragraphs?

Line 317-325: why is this in bold?

Figure 2 shows that P. aeruginosa had the highest degradation, however, was this statistically significant versus the other values? E.g. S. aureus had a 79% which is similar to the 81.49% of P. aeruginosa. Also: the standard deviations seem to be the same for every bacterium?  Where are the controls?

Line 337: is this correct? The degradation is the highest at 20ppm, why not at 5ppm? Also: why are the standard deviations for every tested concentration the same? 

Figure 3: are the obtained results statistically significant? Where are the controls?

Line 353-364: what is the influence of the pH on the absorption spectra of methyl red? The pH alters this but did you take this into account? How do we know the effect seen here is not solely because of a different absorption rate at the used wavelength of 430nm?

Also: does P. aeruginosa survive at a pH <4 ? And >9 ? Where are the controls ? Figure 4 lacks controls.

Fig. 5 and section 3.2.4: its unclear how the 21 day incubation was done if samples were taken every 3 days and the start volume was only 15mL. Also: the cultures were just kept for 21 days straight without addition of new medium? Were the cells not dead/in the starvation phase after several days?

Was the strain used P. aeruginosa a slow growing strain? Normally good growth is seen after 24 hours. Did the addition of the dye slow down the growth ? Why was this not tested? Why was there no growth profile made of the bacterium (comparison regular growth vs growth in medium with the dye)

Figure 6: why are th exact data not under the bars? 

Line 381: repetition, already mentioned in the text.

Line 382: not sure what the point is here as the growth medium also contains a C source (maybe even glucose?) Sentence needs to be rewritten, its not correct now.

Line 393: the higher urea concentrations is toxic for the bacteria? Could it be possible that there is an interaction with the dye and the urea? Why is the best result obtained at 15mg/mL and is the degradation lower at lower concentrations of urea and also lower at higher concentrations? 

Figure 8: very difficult to comprehend that the authors use mg/15mg in the figure and in the text e.g. mention 1000mg/L. Units need to be adjusted and the same units need to be used for the sake of clarity.

Author Response

Reviewer 4

Comment:  Line 23: not sure why 11 is typed in brackets after the word eleven. This can be removed

Response: This mistake has been removed. Thank you

Comment:  Line 25: P. aeruginosa can be used the second (and following) time. No need to write Pseudomonas in full after the first time. Check throughout the rest of the text.

Response: This mistake has been corrected.

Comment:  The first sentence of the introduction (line 50 to 52) is more or less the same as the second sentence (line 52-54). Better to merge both sentences into 1 sentence. Keep both references. 

Response: This has been revised accordingly.  

Comment:  Line 58: without treatment to THE environment. The needs to be added.

Response: The line has been corrected accordingly.

Comment:  Line 63 -  65: check sentence, its the same as the previous sentence and the sentence itself makes no sense. Other life forms in the water vs other living organisms?

Response: This statement has been corrected.

Comment:  Line 68: remove the word induce. 

Response: This word has been removed. Thank you

Comment:  Line 68-70: this sentence is yet again a repetition of previous sentences regarding the disturbance on the aquatic life. Merge with previous sentences and keep references.

Response: This sentence has been revised accordingly.

Comment:  Line 72: put a comma before particularly and a comma after fish. 

Response: Correction has been done.

Comment:  Line 73-74: and also affect ... not affecting. Check sentence in general. Exhibit should be replaced with cause

Response: These mistakes have been removed. Thank you

Comment:  Line 81-84: check sentence and rewrite. Sentence is too long and gibberish near the end.

Response: This sentence has been rewritten and corrected.

Comment:  Line 84-86: check sentence, incorrect. Alternative: Discharge of dyes without any proper treatment can influence aquatic life by blocking light penetration.

Response: This sentence has been corrected.

Comment:  Line 89: wastewater not waste water.

Response: This mistake has been corrected. Thank you

Comment:  Line 93: limitations

Response: This mistake has been corrected. Thank you

Comment:  Line 95: that results IN additional...

Response: This mistake has been corrected. Thank you

Comment:  Line 100: remove "over the time"

Response: This mistake has been corrected. Thank you

Comment:  Line 106-107: literature overview.... yet only 1 reference is given? Are there not more review papers discussing this more in detail?

Response: Reference from literature have been added.

Comment:  Line 107-108: the bacterial body systems? This is not proper English or a proper way to refer to bacteria.

Response: This mistake has been corrected. Thank you

Comment:  Line 109: better reference(s) need to be added, reference 25 is not good enough.

Response: Some other related references have been added.

Comment:  Line 114: studies and not study

Response: This mistake has been corrected. Thank you

Comment:  Line 114-115: not sure what the added value is of adding this extra sentence here about the Ikram et al study? E. coli is mentioned with the other bacteria above. 

Response: Sorry; this extra sentence has been removed.

Comment:  Line 109-115: can be put in a table to make it easier to read.

Response: Table has been added accordingly. Thank You

Comment:  Line 115: bacterial system.? This is not correct English.

Response: This mistake has been corrected. Thank you

Comment:  Line 123: Sentence starts wrong. Better to write e.g.: However, to date only limited studies are available in the literature...  Also: no reference is given here about this statement? 

Response:  These grammatical mistakes were removed and sentence was rephrased accordingly.

Comment:  Line 124 and further: P. aeruginosa, no need to always write Pseudomonas

Response: This mistake has been corrected. Thank you

Comment:  Line 124-126: re-read sentence and change, this is a badly written sentence

Response: The sentence has been rephrased and corrected,

Comment:  Line 126: impcts? Impacts? 

Response:  This mistake has been corrected. Thank you

Comment:  Line 131: metabolites

Response: This mistake has been corrected. Thank you

Comment:  Line 133-137: brand of the chemicals need to be added

Response: The brand of chemicals have been added.

Comment:  Line 142-145: names need to in italic. Also improve sentence by e.g.: the biodegradation efficiency of the following bacteria was tested: .... Also: what are the specific strains? Why are the deposit numbers of the databases not mentioned (where are the strains from?). Some of the bacteria mentioned can be human pathogens. Now it is only mentioned they are from the Abdul Wali Khan University Mardan in Pakistan? But more information should be provided. 

Response: The culture codes /deposit numbers of bacterial strains have been added. No repository of information is available with authors; but the cultures are stored for research purpose only in the department and upon the request they provide the culture to research students/scholars.

Comment:  what is the optimal growth temperatures for these bacterial strains? It is mentioned that growth tests were performed at 37°C but is this the best suited temperature for all the strains?

Response: Normally 37oC is optimal growth temperature. The bacterial colonies were seen visible at this temperature.

Comment:  Line 150: 'about' 0,04g? what is about 0.04?? This is not very scientific.

Response:  This mistake has been corrected. Thank you

Comment:  Line 157: what type of nutrient broth? Brand of the broth needs to be mentioned.

Response: The brand has been mentioned in the text.

Comment:  Line 156-157: re-read sentence, adjust, not proper English

Response: Sentence has been corrected accordingly. Thank you

Comment:  Line 158: glassware not glass wares

Response: This mistake has been corrected. Thank you

Comment:  Line 158: were transferred autoclave? Not proper English. Sentence has to be rewritten. 

Response: This sentence has been rewritten in the manuscript text.

Comment:  Line 158: sterilization at 121°C for 3 hours? Why 3 hours?? 

Response: This is to sterilize the apparatus and media and kill unwanted microbes completely

Comment:  Line 161: not sure what machine stored means here?

Response: Mistake. This mistake has been removed. Thank you.

Comment:  Line 166: re-read sentence, something is wrong. E.g. to each tube 5ml dye was added ... and not what is written now.

Response: The sentence has been rephrased.

Comment:  Line 176: The bacterial strain with the highest percentage...

Response: The said words have been added. Thank you

Comment:   Line 183: by conducting three sets of experiments? I assume you mean the experiments were performed in triplicate?

Response: Yes I mean the experiments were performed in triplicate. Thank you

Comment:  Line 196: Xi is mentioned again?

Response: This repetition has been removed

Comment:  Line 193: unsure why there are two summations shown in the formula at the last summation part

Response: One of the summations expresses effect of linear factors while the other denotes the effect of quadratic function of the process factors.

Comment:  Line 200: not entirely clear what the authors mean with static conditions? Can they explain?

Response: This is because the test tubes were at stationary conditions before investigation.

Comment:  Line 202: space after each comma

Response: Space has been added after each comma in this line.

Comment:  Line 202-203: I assume the authors mean a control was also made for each concentration? A control without bacteria?

Response: Yes; control solution here means without bacteria.

Comment:  205: brand of the centrifuge? Also rewrite sentence, not proper English

Response: Brand of centrifuge (Select Spin Spectra 6C) has been added in text. The sentence has been rephrased.

Comment:  208-214: there is no mentioning of the pH values tested? At what wavelengths was the measurement done? Same as mentioned above? 430nm? If so: is this correct? The absorption spectra of methyl red also depends on the pH of the solution.

Response: The pH values were adjusted using IM HCl and 1M NaOH solutions. By adding minute quantity(microliter) on micropipette in respective tubes; the pH values were tested and adjusted using pH indicator strips purchased from Merck KGaA Darmstadt Germany. These strips have four colors bands and give accurate value of pH as compared to common pH papers

Comment:  Line 201: fourteen is ok, no need to add (14)

Response: Mistake. 14 has been removed. Thank you

Comment:  Line 223-224: samples are now filtered? In the previous paragraphs there is no mentioning of filtering?

Response: Sorry sample filtration was done. This has been mentioned in previous paragraphs. Thank you

Comment:  Line 217: to investigate the effect of temperature effect? This is not proper English

Response: This mistake has been revised.

Comment:  Line 221: mention the tested temperatures in ascending order: 25, 30, 35, 40, 45 and 50°C.

Response: The tested temperatures have been corrected and mentioned in ascending order accordingly.

Comment:  Line 221: are the tested temperatures logical? Will P. aeruginosa survive temperatures of 40 to 50°C ? Why was 37°C not included? Were all the other tests (conditions) performed at 37°C?

Response: From our investigation, there was no significant difference in the temperature range between 35 and 40°C

Comment:  Line 227: not entirely sure what is ment here. The authors mean that the main carbon source was glucose? 

Response: It means that glucose act as an instant source of carbon and energy, it provide extra energy to bacteria as dye have complex structure to degrade.

Comment:  Line 228: different concentrations of glucose were added? But the glucose is stated as a mass not a concentration? What is the final concentration of the samples?

Response: The glucose was added in different concentrations to five test tubes means 5mg/15mL(333.33mg/L), 10mg/15mL(666.66mg/L), 15mg/15mL(1000mg/L), 20mg/15mL(1333.33mg/L) and 25mg/15mL(1666.67 mg/L). We have studied the effect of glucose concentration on dye decolorization in specified volume(15mL) in inoculated test tube. 

Comment:  Line 235-237: not entirely sure what the authors mean here. They mean that the optimal osmotic pressure is important for the bacteria to survive? Or is the salinity also a factor that influences the dye itself? 

Response: Yes; optimal osmotic pressure is important for the bacteria to survive. Optimum salt concentration is required for bacteria. High salt concentration affects bacteria growth.

Comment:  Line 241-246: unsure what is mentioned here. Do the authors mean that the growth of the bacteria was followed for 21 days and every 3 days a sample was taken and measured? If so: how is this possible of the start volume is 15mL (10mL broth, line 161 and 5mL of dye, line 166) and each sample that is taken is 5mL (line 170)

Response: Sorry; I have written test tubes. It is test tube in line. Normal test tube has 15-20 mL capacity. The effect of time on biodegradation experiment was conducted in large sized test tube having 30 mL media plus 15 mL dye solution. After every 3 days interval 5mL sample was taken, centrifuged then filtered and its absorbance was recorded by UV Visible Spectrophotometer. The rest of the experiments were performed in test tube having 15mL capacity (10mL media plus 5mL dye)

Comment:  Line 248-253: why was urea added? The bacteria were grown in nutrient (bacterial) broth, which provides all the necessary nutrients? 

Response: It is extra source of nitrogen. Urea provide nitrogen that is required for DNA and protein synthesis of bacteria.

Comment:  Line 256-257: incorrect English, needs to be rephrased.

Response: The English of this line has been corrected in text.

Comment:  Line 258: I assume the 1mg/15mL is the final concentration used? So 1mg of each tested substance was added?

Response: Yes, 1mg of each redox mediator was added.

Comment:  Line 273: was crushed, unsure what the authors mean with crushed

Response: Crush mean  to form a homogeneous mixture.

Comment:  Line 274: why centrifuged for 30 minutes here compared to 10 minutes in previous paragraphs?

Response: Mistake. It has been corrected in text.  

Comment:  Line 317-325: why is this in bold?

Response: This mistake has been corrected.

Comment:  Figure 2 shows that P. aeruginosa had the highest degradation, however, was this statistically significant versus the other values? E.g. S. aureus had a 79% which is similar to the 81.49% of P. aeruginosa. Also: the standard deviations seem to be the same for every bacterium?  Where are the controls?

Reviewer: The percentage degradation value (81.49) is quite high than 79. It is a big difference regarding decolorization.

Comment:  Line 337: is this correct? The degradation is the highest at 20ppm, why not at 5ppm? Also: why are the standard deviations for every tested concentration the same? 

Response: Substrate concentration is important factor. At low concentration of Dye (i.e 5ppm)  the bacterial enzymes like azoreductases are not fully saturated hence enzymatic activity is low and degradation is minimum. At 20ppm the enzymes are fully saturated and degradation activity is maximum. Beyond 20ppm degradation declines as high concentration results in inactivation of active sites of enzymes.  Standard deviation (error bars) is different in each case. It is typographical mistake and has been removed.

Comment:  Figure 3: are the obtained results statistically significant? Where are the controls?

Response: Yes, these results are significant statistically.

Comment:  Line 353-364: what is the influence of the pH on the absorption spectra of methyl red? The pH alters this but did you take this into account? How do we know the effect seen here is not solely because of a different absorption rate at the used wavelength of 430nm?

Response: The effect of pH was reflected on the degradation efficiency of the bacteria based on linear optimization and also as a process variable which determined the optimum condition of the factors and their interaction using the central composite design.  

Comment:  Also: does P. aeruginosa survive at a pH <4 ? And >9 ? Where are the controls ? Figure 4 lacks controls.

Response: At extreme conditions of pH both in acidic and alkaline medium, bacteria growth is inhibited.

Reviewer: Fig. 5 and section 3.2.4: its unclear how the 21 day incubation was done if samples were taken every 3 days and the start volume was only 15mL.

Response: Response: Sorry; I have written test tubes. It is test tube in line. Normal test tube has 15-20mL capacity.  The effect of time on biodegradation experiment was conducted in large sized test tube having 30 mL media plus 15 mL dye solution. After every 3 days interval 5mL sample was taken, centrifuged then filtered and its absorbance was recorded by UV Visible Spectrophotometer. The rest of the experiments were performed in test tube having 15mL capacity (10mL media plus 5mL dye).

Comment:  Also: the cultures were just kept for 21 days straight without addition of new medium? Were the cells not dead/in the starvation phase after several days?

Response: Initially the biomass production was rapid. However, after 3 days of incubation the bacterial colonies were found to compete for nutrients hence the degradation capacity of bacteria decreased. The metabolic activities and release of enzymes also stops with the passage of time so degradation decreases. May be some bacteria die in the starvation phase due to non-availability of nutrient.

Comment:  Was the strain used P. aeruginosa a slow growing strain? Normally good growth is seen after 24 hours. Did the addition of the dye slow down the growth? Why was this not tested? Why was there no growth profile made of the bacterium (comparison regular growth vs growth in medium with the dye)

Response: The culture of P. aeruginosa was fully grown and visible after 24 hours. However, after addition of dye solution its growth was not checked. Growth may be affected by addition of dye.

Comment:  Figure 6: why are the exact data not under the bars? 

Response: The standard deviation of the percent degradation is indicated in the error bars as illustrated in the Figures.

Comment:  Line 381: repetition, already mentioned in the text.

Response: This repetition has been removed

Comment:  Line 382: not sure what the point is here as the growth medium also contains a C source (maybe even glucose?) Sentence needs to be rewritten, its not correct now.

Response: Glucose act as instant source of energy and provide carbon to bacteria.

Comment:  Line 393: the higher urea concentrations is toxic for the bacteria? Could it be possible that there is an interaction with the dye and the urea? Why is the best result obtained at 15mg/mL and is the degradation lower at lower concentrations of urea and also lower at higher concentrations? 

Response: The urea has interaction with bacteria and not the dye. In our experiment 15mg/L is optimum concentration for bacteria to degrade the dye; below and above this concentration the degradation activity of bacteria decreases. As urea is instant source of N for bacteria but high concentration may be toxic and lower concentration does not initiate the degradation reaction properly.  

Comment: Figure 8: very difficult to comprehend that the authors use mg/15mg in the figure and in the text e.g. mention 1000mg/L. Units need to be adjusted and the same units need to be used for the sake of clarity.

Response: The concentration units have been corrected and presented in more normalized fashion i.e mg/L in the whole manuscript. Thank you

Reviewer 5 Report

The authors analyze optimized process conditions for biodegradation of Azo dye Methyl Red by Pseudomonas aeruginosa. 

Specific comments to the authors:

Abstract: Needs to be more specific.line 20, 21 and 22 should provide specific information.

Line 33-36  Reframe the sentence.

Line 40-41 check English throughout. 

Introduction: Vague information in the beginning till line 87. Cite recent references with specific information. English should be improved.

Materials and methods: Do not present structure in the materials and methods section. Write the names of the bacteria in italics!

Results: For the bar diagrams, use a different colour. Only a specific temperature and pH range was mentioned. Mention the entire pH and temperature profile which was taken into consideration.For the GC-MS analysis the authors must mention how the concentration of the contaminants were calculated.

Overall, the manuscript needs a complete overall in terms of presentation of data and the scientific language used.

Author Response

Reviewer 5

The authors analyze optimized process conditions for biodegradation of Azo dye Methyl Red by Pseudomonas aeruginosa. 

Specific comments to the authors:

Comment:  Abstract: Needs to be more specific. line 20, 21 and 22 should provide specific information.

Response: The lines have been revised accordingly.

Comment:  Line 33-36 Reframe the sentence.

Response: These sentences has been revised.

Comment:  Line 40-41 check English throughout. 

Response: The abstract has been revised concisely.

Comment:  Introduction: Vague information in the beginning till line 87. Cite recent references with specific information. English should be improved.

Response: This has been revised. Also, the references has been updated.

Comment:  Materials and methods: Do not present structure in the materials and methods section. Write the names of the bacteria in italics!

Response: The structure of dye has been removed from this section. Moreover, the names of bacterial strain have been written in italics and provided with culture codes.

Comment:  Results: For the bar diagrams, use a different color. Only a specific temperature and pH range was mentioned. Mention the entire pH and temperature profile which was taken into consideration..

Response: The error bars are now visible. Due to high resolution the error bars were not visible The pH range was mentioned from 1-14 while the temperature range was 25,30,35,40,45 and 500C. At pH 9 and 400C the percentage degradation rates were maximum so these were considered in further experiments.

Comment: For the GC-MS analysis the authors must mention how the concentration of the contaminants were calculated

Response: The concentration of contaminants/metabolites were calculated by comparing the peak area and retention times in the sample with the peak area of the standard reported compounds in literature.

This has been mentioned in the manuscript.

Comment:  Overall, the manuscript needs a complete overall in terms of presentation of data and the scientific language used.

Response: The manuscript has been revised accordingly.

Round 2

Reviewer 4 Report

Dear authors,

the paper has improved, however there are still some major issues to be addressed. Especially regarding the controls. Figures need to be improved as well. Text need serieus copy-editing. Material and methods needs to be improved and streamlined.

1° It is mentioned that you autoclaved for 3 hours but do you mean 3 hours for the total run or 3 hours at 121°C? If you autoclave for 3 hours at 121°C the medium will be destroyed.

2° You mention the following: "At extreme conditions of pH both in acidic and alkaline medium, bacteria growth is inhibited. " It is not clear to me why you do not address this in the text. If the bacteria did not grow or were inhibited at extreme high/low pH values, this should be mentioned in the results and discussion section. Also: there should be a control shown of the bacterial growth in these low/high pH values. Perhaps the % degradation at low/high pH values is not due to the bacterium. If the bacterium does not survive in low/high pH values, than what is causing the degradation?

3° a similar remark on the temperature: where is the control growth in 50°C. Is the bacterium actually growing at 50°C? Why is the control not shown?

4° In general: there should be more controls shown. They can be shown in a supplementary file. Readers need to know what the growth of the bacterium is in these conditions without the dye. If you do not show the control readers do not know what the effect is of the bacterium on the dye. Example: if the bacterium does NOT grow at pH 1 and yet you have 37.16% degradation in this condition, than this degradation comes from natural degradation and NOT from the bacterium.

5° The statistical significance needs to be highlighted on the graphs. Example: figure 2= at 10PPM the %degradation is 66.13, while at 15ppm it is 66.39. I assume this is not a significant difference? Add the differences at the top of each bar to show which bars are statistically significant different from each other.

6° How come that all of a sudden there are more authors on the paper ?

7° Why was the dye added AFTER bacteria already grew 24 hours?

8° Material and methods needs to be improved. It is often very confusing. Eg: you mention that the bacterium was grown for 24 hours before the dye was added in 2.5, yet later on in 2.7.3 you write bacteria and dye was added to the testtubes, here it seems it was added together. It is very confusing. On top of this, sometimes you mention adding the dye, sometimes you don't , eg. 2.7.2 does not mention anything about the dye was added. This needs to be streamlined. Suggested is to make a general paragraph on how the experiments were done (describe adding the bacterium, the dye etc) and then in the subsequent paragraphs you can briefly address how the parameters eg temperature was tested.

9° Figure 20, was this figure made by the authors? In case not: what is the original source? Same for fig 19

10° You mention in line 401 that an additional carbon source is needed because the dyes are not easy to be degraded, however, does the nutrient broth not contain a carbon source? Also: do the bacteria use the dye as a carbon source? Because now it seems that they do. The way the text is written implies the bacterium using the dye as a carbon source, which is not necessarily true? 

11°: please adjust the figures in such a way we can see the exact value below each graph. I asked this before, but it was misunderstood. The point I am trying to make is, for example, figure 6,: based on the graph I can not see what the glucose concentration was for each bar since the bars are located between numbers. The first bar is located between 200 and 400 PPM... the second one between 600 and 800 ppm... Adjust the graphs so that you see the EXACT number below each bar. See for a good example figure 4: there the exact temperature is below each bar. 

12° Line 521: optimal degradation (not optimum) was with 19.4 days of incubation? How is this possible? You mention in the text that the maximum of %degradation was already reached after 3 days. ? So how come its 19.4 days? Line 677: optimal condition is 3 days? But you mentioned 19.4 before with the statistical test? So what is it? 3 days? 19.4 days?

13°: sometimes you speak about maximum decolorization, but in the figures you mention on the y-axis the % degradation. It is the same (as you mention in line 178, but you should be more consistent in using the proper terminology. If the figures show % degradation, then also mention degradation in the text. 

14° Line 203-205: this needs to be rephrased. I assume you mean the bacteria were already in stationary phase? Also: you mean in general that the bacterium was allowed to grow for 24 hours and then the dye was added ? Like you mention in 2.5? This needs to be more consistent, see also remark 8.

15° Line 248: large size test tube? this means nothing, mention the volumes used. 

16° how come the standard deviations in most of the figures are the same for most bars? Example figure 2: the standard deviations seem all the same?

17° Perhaps it is better to change the color (blue) in the figures to a more neutral (better visible) color such as grey. 

18° Check line 481 , figure 10, something went wrong with the caption of the figure.

19° see my previous comment:

Comment: Line 353-364: what is the influence of the pH on the absorption spectra of methyl red? The pH alters this but did you take this into account? How do we know the effect seen here is not solely because of a different absorption rate at the used wavelength of 430nm?

Response: The effect of pH was reflected on the degradation efficiency of the bacteria based on linear optimization and also as a process variable which determined the optimum condition of the factors and their interaction using the central composite design.

I am not sure my question was misunderstood. My question is: methyl red is a pH indicator, so depending on the pH it will change color. If you changed the pH of your growth medium for the bacterium to grow, then you also change the color of the methyl red. How do you take this into account? Are the samples with a low pH not different in color compared to the ones with a more neutral pH? I assume the control samples were used as a blanc to measure the OD ? 

20° why is there no control shown in the graphs that shows the degradation of the dye without addition of the bacteria? As long as this is not shown, we do not know what the true effect of the bacterium is ! Perhaps the dye degrades naturally for X % ...

21° Line 512-513; not sure this is a correct statement. The pH facilitates the interaction with the cell wall? And this comes because of increased incubation times? Do the authors simple mean that with a prolonged incubation time the dye degradation increases? And that the pH is also an essential parameters in the degradation of the dye?

21° Figure 10: what does it show? On the left it is a sample at the start of incubation? Or ? On the right, these are samples after incubation? I can see precipitation at the bottom of the tubes, are these the bacteria?

Minor issues with the text:

Line 54: over few decades ago is not correct English, best to change into: during the last decades the environmental problems arising from the textile industry have gained more attention. 

Line 65-66: no need to say 'other aquatic species', this is part of the aquatic life. Just remove other aquatic species.

Line 115: 'bacterial system has' => bacterial system is not really a correct way to describe what you want to state. Better to change it into: Bacteria have multiple enzymes such as... 

Line 125: limited studies to date, do you have a reference on this? Which papers are already discussing this? perhaps they can be added as a reference or a review paper that states that until this day only limited studies are available.

Line 143-150: check the spaces between the commas and the ATCC numbers, do this throughout the text as often a space is forgotten between words/letters etc

Line 152: 40ppm should be 40 ppm ... Check this throughout the text, see also comment above, you need to be more cautious when writing.

Line 162: were kept in inside ... Remove the 'in' , inside already means in.

Line 162: remove 'machine' after autoclave. Its redudant.

Line 174-175: sentence needs to be changed, now it states that the bacterial mass was used for further tets. Change into: The bacterial cell mass was separated from the supernatant and the supernatant was subsequently used to conduct...

Line 189: .. efficiency of bacterial strain.. not correct English. Change into: the degradation efficiency by P. aeruginosa ... or if you do not want to mention the bacterium by name you can write: efficiency by the selected bacterium... 

Libe 216: Control solutions were also prepared as a reference. Prepared from a reference is incorrect.

Line 338: this sentence is all wrong. Also the conclusion is a bit strange. There is no further degradation anymore after 3 days because the bacteria are most likely dead and the degradation maximum was already reached at day 3. Also re-write this sentence to something like: After 3 days of incubation a maximum in degradation was observed. No significant increase in degradation was seen after these 3 days. This is most likely due to reaching the stationary and death phase. 

Line 401: there AN additional source of...

Line 437: was able TO decolorize..

Line 451: seems to be a different font or size used? Check the 'it has been reported' part, this seems smaller that then rest of the text

Line 508-510: please put reference 70 behind this sentence: The degradation capacity of bacteria strain greatly depends on pH, microbial growth and the conversion of complex substances into a simpler form " 

Line 676: not sure what is ment with this sentence: "the chosen bacterial strain..." ? You used this strain because it showed the highest potential in your preliminary test.

Line 677: optimal condition is 3 days? But you mentioned 19.4 before with the statistical test?

These are just a few examples: serious copy-editing of the text is needed.

Author Response

Reviewer 4

Dear authors,

Comment: The paper has improved, however there are still some major issues to be addressed. Especially regarding the controls. Figures need to be improved as well. Text need serieus copy-editing. Material and methods need to be improved and streamlined.

Response: The manuscript text has been revised and improved accordingly.

Comment:  It is mentioned that you autoclaved for 3 hours but do you mean 3 hours for the total run or 3 hours at 121°C? If you autoclave for 3 hours at 121°C the medium will be destroyed.

Response:  We autoclaved 3 hours for the total run not 3 hours at 121°C as the medium can be destroyed if we autoclave it for 3 hours at 121°C. It is process time otherwise sterilization is only for 15 min of the total cycle including initial warm up, then sterilization of 15 min, then cooling and fan operation etc.

Comment:  You mention the following: "At extreme conditions of pH both in acidic and alkaline medium, bacteria growth is inhibited. " It is not clear to me why you do not address this in the text. If the bacteria did not grow or were inhibited at extreme high/low pH values, this should be mentioned in the results and discussion section. Also: there should be a control shown of the bacterial growth in these low/high pH values. Perhaps the % degradation at low/high pH values is not due to the bacterium. If the bacterium does not survive in low/high pH values, than what is causing the degradation?

Response: Worthy reviewer it does not mean that degradation stopped. At extreme conditions of pH, both acidic and alkaline bacterial growth and enzymatic activity was affected so its dye degradation potential was also affected. We are interested in finding out the optimum pH rather than sorting out other reason of degrading in this set of experiment. The statement was accordingly inserted in the respective section.

Comment: a similar remark on the temperature: where is the control growth in 50°C. Is the bacterium actually growing at 50°C? Why is the control not shown?

Response: Thank you referee, growth was achieved at 50°C. Above optimum temperature, steady decrease occurs in decolorization which may probably be due to the denaturation of bacterial enzymes [52]. At high temperature enzymes like azo reductase, laccases were affected so degradation decreased. Worthy reviewer, in evaluating temp effect 50C is one set of temperature at we observed decrease in the rate of degradation probably due to the mentioned reason of enzyme deactivation. Since our optimum temp was below this then we were not interested whether it is temp effect of bacterial effect. If we have performed all these experiment at 50C then control evaluation was mandatory but it is just minor experiment in selecting optimal conditions. Your point is valid however, if controls are run for each experiment, then we will loss the aim of our study. Also, in most of the literature study control solutions degradation is not given for each set of experiment rather one control is given. At optimal condition control are already shown there where the color of control solution in tubes did not change with respect to bacteria treated solution.

Comment: In general: there should be more controls shown. They can be shown in a supplementary file. Readers need to know what the growth of the bacterium is in these conditions without the dye. If you do not show the control readers do not know what the effect is of the bacterium on the dye. Example: if the bacterium does NOT grow at pH 1 and yet you have 37.16% degradation in this condition, than this degradation comes from natural degradation and NOT from the bacterium.

Response: Worthy Reviewer, as I pointed out that we are interested defining optimum conditions and it is quite difficult to run control for each set i.e run control for individual pH where tested samples are more than 8. It means I have to run 8 control similar will be the case with each temp. Worthy reviewer, please check literature control is only run for the sample where you are getting the desired results. It is a good point to record the degradation in control solutions for each point but it will increase the labor of a researcher many folds. We were interested in defining optimal condition as I mentioned before and for optimal sample control are already there. At 50C for example I  have less degradation then I have discarded this temp as it is not fulfilling my desired criteria. Now I have interest in this temp therefore I discarded. If my research paper title was bacterial degradation at 50C then it was mandatory for me to run control. However, here it is not my interest therefore we have not studied.   

Comment: The statistical significance needs to be highlighted on the graphs. Example: figure 2= at 10ppm the % degradation is 66.13, while at 15ppm it is 66.39. I assume this is not a significant difference? Add the differences at the top of each bar to show which bars are statistically significant different from each other.

Response: The degradation at 10 ppm 66.13 ± 0.985 while at 15 ppm it is 66.39± 0.921. Although the difference is not too much but statistically the percent degradation is more at 15ppm.  

Comment: How come that all of a sudden there are more authors on the paper?

Response: The authors added contributed in the manuscript. This was a mistake in the previous manuscript not to include these co-authors. Also according to the journal policy author addition at latter stages is permissible. Please check MDPI policy.

Comment: Why was the dye added AFTER bacteria already grew 24 hours?

Response: Worthy reviewer, for degradation bacterial cells are needed which are very few in initial inoculum added. Therefore, to increase their number, Firstly, bacteria should grow in media after which dye solution should be added to decolorize the dye efficiently.

Comment: Material and methods needs to be improved. It is often very confusing. Eg: you mention that the bacterium was grown for 24 hours before the dye was added in 2.5, yet later on in 2.7.3 you write bacteria and dye was added to the test tubes, here it seems it was added together. It is very confusing. On top of this, sometimes you mention adding the dye, sometimes you don't , eg. 2.7.2 does not mention anything about the dye was added. This needs to be streamlined. Suggested is to make a general paragraph on how the experiments were done (describe adding the bacterium, the dye etc) and then in the subsequent paragraphs you can briefly address how the parameters eg temperature was tested.

Response: Material and method section has been revised accordingly.

Comment: Figure 20, was this figure made by the authors? In case not: what is the original source? Same for fig 19

Response: Yes, these figures has been made by the authors.

Comment: You mention in line 401 that an additional carbon source is needed because the dyes are not easy to be degraded, however, does the nutrient broth not contain a carbon source? Also: do the bacteria use the dye as a carbon source? Because now it seems that they do. The way the text is written implies the bacterium using the dye as a carbon source, which is not necessarily true? 

Response: Nutrient broth also contain carbon but additional carbon source could increase the degradation efficiency of bacteria. According to literature in some conditions there is negative impact of increasing sugar concentration on the bacterial metabolic pathway to catabolize sugar; so increase in glucose concentration would decrease the degradation efficiency of bacteria. Worthy reviewer, dyes in effluent also contain a number of organic and inorganic waste products, if extra carbon source or salt have negative impact then it is needed to clarify them for large scale applications. You know it better that effluent usually mixes with house sewerage line which abundantly contain such constituent, so their impact on degradation is mandatory to be evaluated. For real sample applications such things are evaluated.

Comment: please adjust the figures in such a way we can see the exact value below each graph. I asked this before, but it was misunderstood. The point I am trying to make is, for example, figure 6,: based on the graph I can not see what the glucose concentration was for each bar since the bars are located between numbers. The first bar is located between 200 and 400 PPM... the second one between 600 and 800 ppm... Adjust the graphs so that you see the EXACT number below each bar. See for a good example figure 4: there the exact temperature is below each bar.

Response: The graphs has been adjusted with exact number below accordingly. Thank you

Comment: Line 521: optimal degradation (not optimum) was with 19.4 days of incubation? How is this possible? You mention in the text that the maximum of %degradation was already reached after 3 days. ? So how come its 19.4 days? Line 677: optimal condition is 3 days? But you mentioned 19.4 before with the statistical test? So what is it? 3 days? 19.4 days?

Response: Thank you referee, there are two optimization stages in our study. The first phase of optimization is a one-time factor optimization that is based on the effect of single factor on the degradation of dye. The effect of time as a single factor on the degradation of dye occurred after 3 days (Section 3.2.4). However, the second phase of our study focused on the combined effect of the interaction of four significant parameters influencing dye degradation using the second order polynomial model of the central composite design of the response surface methodology. These parameters were pH, incubation period, and temperature and dye concentration. Thus, with the consideration of the four-process conditions being optimized concurrently, the optimal degradation was enhanced when pH was 10 within incubation period of 19.4 days at temperature of 31.54 oC dye concentration of 17.24 ppm.  

 Comment: sometimes you speak about maximum decolorization, but in the figures you mention on the y-axis the % degradation. It is the same (as you mention in line 178, but you should be more consistent in using the proper terminology. If the figures show % degradation, then also mention degradation in the text. 

Response: Worthy reviewer here is maximum degradation. The mistake has been corrected in the text. However, in some cases the chromophoric group just simple undergo protonation or deprotonation where the term decolorization is preferred rather than degradation.  

Comment: Line 203-205: this needs to be rephrased. I assume you mean the bacteria were already in stationary phase? Also: you mean in general that the bacterium was allowed to grow for 24 hours and then the dye was added? Like you mention in 2.5? This needs to be more consistent, see also remark 8.

Response: These lines has been rephrased accordingly. Thank you.

Comment: Line 248: large size test tube? this means nothing, mention the volumes used.

Response: The volume of the nutrient broth media was 30 mL while that of dye was 15 mL so total volume 45 mL. This has been corrected and mentioned in the the manuscript.

Comment: how come the standard deviations in most of the figures are the same for most bars? Example figure 2: the standard deviations seem all the same?

Response: In figure 2 there is difference in standard deviation values but the difference is too small so the error bars looks like same.

Comment: Perhaps it is better to change the color (blue) in the figures to a more neutral (better visible) color such as grey. 

Response: The color of the figures has been changed from blue to grey. Thank you

Comment: Check line 481, figure 10, something went wrong with the caption of the figure.

Response: This caption has been improved.

Comment: Line 353-364: what is the influence of the pH on the absorption spectra of methyl red? The pH alters this but did you take this into account? How do we know the effect seen here is not solely because of a different absorption rate at the used wavelength of 430nm?

Response: The effect of pH was reflected on the degradation efficiency of the bacteria based on linear optimization and also as a process variable which determined the optimum condition of the factors and their interaction using the central composite design.

Comment: I am not sure my question was misunderstood. My question is: methyl red is a pH indicator, so depending on the pH it will change color. If you changed the pH of your growth medium for the bacterium to grow, then you also change the color of the methyl red. How do you take this into account? Are the samples with a low pH not different in color compared to the ones with a more neutral pH? I assume the control samples were used as a balance to measure the OD? 

Response: Worthy reviewer, we are interested in degradation rather than decolorization. The effect of acid and base pH on methyl red is protonation and deprotonation where the term decolorization is preferred rather than degradation.  Degradation is some different phenomenon. We know that; methyl red is a pH indicator as well. If the methyl red indicator is added to an acid solution with pH less than 4.4 the indicator in solution takes a red colour If the methyl red indicator is added to a solution with pH greater than 6.2, the indicator in solution takes a yellow/orange colour. The colour of the control sample was retained because the sample pH was in alkaline region. Therefore, if the methyl red indicator is added to a NaOH solution which is basic with pH much higher than 7, then the solution will have a yellow/orange colour. The optimized treated samples adjusted at alkaline pH were decolorized and their color changed.  

Comment: why is there no control shown in the graphs that shows the degradation of the dye without addition of the bacteria? As long as this is not shown, we do not know what the true effect of the bacterium is ! Perhaps the dye degrades naturally for X % ...

Response:  Worthy Reviewer, a solution without bacteria is control solution by itself. What will a control for it, we do not about a control for a control solution.   

Comment: Line 512-513; not sure this is a correct statement. The pH facilitates the interaction with the cell wall? And this comes because of increased incubation times? Do the authors simple mean that with a prolonged incubation time the dye degradation increases? And that the pH is also an essential parameters in the degradation of the dye?

Response: Worthy reviewer; the cell membrane permeation efficiency is affected by pH, it has been identified as the limiting step in the breakdown of dye by bacteria [71]. So pH is also one of the important essential parameters in the dye degradation. When incubation time increase the pH facilitates the interaction of methyl red to the cell membrane of the bacterial strain, thereby resulting in increase in the dye degradation. Time effect is separately elaborated in other section.

Comment: Figure 10: what does it show? On the left it is a sample at the start of incubation? Or ? On the right, these are samples after incubation? I can see precipitation at the bottom of the tubes, are these the bacteria?

Response: Figure 10 shows Color of methyl red dye (a) before P. aeruginosa treatment (b) after treatment of P. aeruginosa. Yes at the bottom precipitation of the dye degraded mixture and bacterial cell mass.

Minor issues with the text:

Comment: Line 54: over few decades ago is not correct English, best to change into: during the last decades the environmental problems arising from the textile industry have gained more attention. 

Response: The sentence has been revised accordingly.

Comment: Line 65-66: no need to say 'other aquatic species', this is part of the aquatic life. Just remove other aquatic species.

Response: This mistake has been corrected accordingly

Comment: Line 115: 'bacterial system has' => bacterial system is not really a correct way to describe what you want to state. Better to change it into: Bacteria have multiple enzymes such as... 

Response: This mistake has been corrected accordingly

Comment: Line 125: limited studies to date, do you have a reference on this? Which papers are already discussing this? perhaps they can be added as a reference or a review paper that states that until this day only limited studies are available.

Response: The reference has been added in text.

Comment: Line 143-150: check the spaces between the commas and the ATCC numbers, do this throughout the text as often a space is forgotten between words/letters etc

Response: This mistake has been corrected accordingly.

Comment: Line 152: 40ppm should be 40 ppm ... Check this throughout the text, see also comment above, you need to be more cautious when writing.

Response: This mistake has been corrected throughout. Thank you

Comment: Line 162: were kept in inside ... Remove the 'in' , inside already means in.

Response: This mistake has been corrected. Thank you

Comment: Line 162: remove 'machine' after autoclave.

Response: This mistake has been corrected. Thank you

Line 174-175: sentence needs to be changed, now it states that the bacterial mass was used for further tets. Change into: The bacterial cell mass was separated from the supernatant and the supernatant was subsequently used to conduct...

Response: The sentence has been revised accordingly.

Comment: Line 189: efficiency of bacterial strain. not correct English. Change into: the degradation efficiency by P. aeruginosa ... or if you do not want to mention the bacterium by name you can write: efficiency by the selected bacterium... 

Response: This mistake has been correcte. Thank you

Comment: Libe 216: Control solutions were also prepared as a reference. Prepared from a reference is incorrect.

Response: This mistake has been corrected. Thank you

Comment: Line 338: this sentence is all wrong. Also the conclusion is a bit strange. There is no further degradation anymore after 3 days because the bacteria are most likely dead and the degradation maximum was already reached at day 3. Also re-write this sentence to something like: After 3 days of incubation a maximum in degradation was observed. No significant increase in degradation was seen after these 3 days. This is most likely due to reaching the stationary and death phase. 

Response: These sentences has been revised accordingly.

Comment: Line 401: there AN additional source of...

Response: This mistake has been corrected. Thank you

Comment: Line 437: was able TO decolorize.

Response: This mistake has been corrected. Thank you

Comment: Line 451: seems to be a different font or size used? Check the 'it has been reported' part, this seems smaller that then rest of the text

Response: The font size has been corrected accordingly. Thank you

Comment: Line 508-510: please put reference 70 behind this sentence: The degradation capacity of bacteria strain greatly depends on pH, microbial growth and the conversion of complex substances into a simpler form " 

Response: The reference has been placed at correct position.

Comment: Line 676: not sure what is ment with this sentence: "the chosen bacterial strain..." ? You used this strain because it showed the highest potential in your preliminary test.

Response: This mistake has been corrected. Thank you

Comment: Line 677: optimal condition is 3 days? But you mentioned 19.4 before with the statistical test?

Response: The first stage optimization which considers one parameter under fixed condition of other parameters resulted in optimal degradation within 3 days.  However, the second phase of our study focused on the combined effect of the interaction of four significant parameters influencing dye degradation using the second order polynomial model of the central composite design of the response surface methodology. These parameters were pH, incubation period, and temperature and dye concentration. Thus, with the consideration of the four-process conditions being optimized concurrently, the optimal degradation was enhanced when pH was 10 within incubation period of 19.4 days at temperature of 31.54 oC dye concentration of 17.24 ppm. 

Comment: These are just a few examples: serious copy-editing of the text is needed.

Response: The paper has been revised and edited accordingly

Reviewer 5 Report

The manuscript has been modified according to the comments. It is now acceptable in its present form.

Author Response

Reviewer 5

Comment: The manuscript has been modified according to the comments. It is now acceptable in its present form.

Response: Worthy Reviewer Thank you very much for your appreciation. Your valuable comments have improved the quality of manuscript.